# User-item fairness tradeoffs in recommendations

**Sophie Greenwood**
Computer Science
Cornell Tech

**Sudalakshmee Chiniah**
Operations Research and Information Engineering
Cornell Tech

**Nikhil Garg**
Operations Research and Information Engineering
Cornell Tech

## Abstract

In the basic recommendation paradigm, the most (predicted) relevant item is recommended to each user. This may result in some items receiving lower exposure than they "should"; to counter this, several algorithmic approaches have been developed to ensure *item fairness*. These approaches necessarily degrade recommendations for some users to improve outcomes for items, leading to *user fairness* concerns. In turn, a recent line of work has focused on developing algorithms for multi-sided fairness, to jointly optimize user fairness, item fairness, and overall recommendation quality. This induces the question: *what is the tradeoff between these objectives, and what are the characteristics of (multi-objective) optimal solutions?* Theoretically, we develop a model of recommendations with user and item fairness objectives and characterize the solutions of fairness-constrained optimization. We identify two phenomena: (a) when user preferences are diverse, there is "free" item and user fairness; and (b) users whose preferences are misestimated can be *especially* disadvantaged by item fairness constraints. Empirically, we prototype a recommendation system for preprints on arXiv and implement our framework, measuring the phenomena in practice and showing how these phenomena inform the *design* of markets with recommendation systems-intermediated matching.

## 1 Introduction

Recommendation systems are employed throughout modern online platforms to suggest *items* (media, songs, books, products, or jobs) to *users* (viewers, listeners, readers, consumers, or job seekers). The platform learns user preferences and shows each user personalized recommendations. One recommendation paradigm is to simply show the user the items they most prefer. However, this approach may result in disparately poor outcomes for some items, which may not be most preferred by any user [44]. For example, in our empirical application in prototyping a recommender system for arXiv preprints, we find that on average more than 47% of papers have less than a $0.0001\%$ probability of being recommended to any user, even when the number of users and items are the same. Thus, many algorithmic techniques have been proposed to improve item fairness in recommendation [3, 35, 48, 50]. However, by not solely optimizing for user engagement, these techniques impose a cost both to overall recommendation quality [3] and especially for some individual users more than others.

Accordingly, algorithms have recently been introduced to address the problem of *two-sided fairness* (or *multi-sided fairness*), in which the platform aims to balance user fairness, item fairness, and overall recommendation quality [11, 12, 47]. These algorithms formalize the desired balance in terms of an optimization problem – for example, maximizing the difference between overall recommendation quality and unfairness penalties [11], or the overall recommendation quality subject to fairness

38th Conference on Neural Information Processing Systems (NeurIPS 2024).

constraints [12]. The relative importance of user and item fairness is described by the relative strength of the respective unfairness penalties or slack in the fairness constraints.

However, an open question is, *what are the implications of such algorithms on the recommendations, i.e., what do multi-sided constraints do, and what is the price of (multi-sided) fairness?* More specifically, *are there settings in which we can simultaneously maximize all objectives, "for free?," as opposed to there being large tradeoffs as commonly emphasized? Are some users or items – for example, those new to the platform – more affected than others? How do the answers to the above questions depend on the context and, in real-world settings, do "fairness" constraints substantially affect recommendation characteristics?* Such real-world considerations are essential for platform designers to understand. In fact, a recent survey and critique of the fair ranking literature advocated for such grounded analyses to understand algorithmic implications and tradeoffs [40], as opposed to black-box deployment of fairness algorithms.

Answering such questions is challenging. Theoretically, multi-sided fairness is cast as an optimization problem, and conceptually characterizing optimization solutions is often intractable. For example, given an arbitrary utility matrix and a constraint on the exposure provided to each item, it is not tractable to calculate recommendations in closed form: the solution depends on global structure, users may not receive their most preferred items, and items may not be recommended to the users who most prefer them. Then, once phenomena are theoretically identified, empirically verifying phenomena requires specifying a recommendation setting and measuring user-item utilities as a function of their characteristics. Given these challenges, our contributions are as follows.

**Theoretical framework to characterize solutions of multi-sided fair recommendations.** We formulate a concave optimization problem in which user fairness is formalized as an objective on the minimum normalized utility provided to each user – and item fairness constraints determine the problem's feasible region, through the solutions of another concave optimization. This formulation qualitatively captures standard multi-objective approaches for user-item fairness [5, 12]. We show that when item fairness constraints are maximal, the solutions to this optimization problem (recommendation probabilities for each user) have a sparse structure that can be characterized as a function of the problem inputs (e.g., estimated utilities for each user-item pair, or slack given in the item fairness constraint).

**Conceptual insights on the price of fairness.** We use our theoretical framework to characterize user-item fairness tradeoffs. We identify two phenomena: (a) "free fairness" as a function of user preference diversity: if users have sufficiently diverse preferences, imposing item fairness constraints can have large benefits to individual items with little cost to users, i.e., there is a small *price of fairness*. (b) "Reinforced disparate effects" due to preference uncertainty. Of course, users for whom the platform has poor preference estimation (e.g., "cold start" users on whom the platform has no data) typically receive more inaccurate recommendations; we show that this effect may be *worsened* with item fairness constraints, in a worst case sense: when a user's preferences are uncertain, item-fair recommendation algorithms will recommend them the globally least preferred items – *even when attempting to maximize the minimum user utility*.

**Empirical measurement.** Finally, we use real data to prototype a recommendation engine for new arXiv preprints and use this system to measure the above phenomena in practice. For example, we find that more homogeneous groups of users have steeper user-item fairness tradeoffs – as theoretically predicted, diverse user preferences decrease the price of item fairness. Furthermore, we find that the "price of misestimation" is high (users for whom less training data is available receive poor recommendations), but on average item fairness constraints do not increase this cost.

Putting things together, we show that the real-world effects of user-item recommendation fairness constraints heavily depend on the empirical context. In some cases, "item fairness" comes for free, with little cost to users. In others, deploying such an algorithm may lead to especially poor recommendations for some users, in ways that cannot be mechanically addressed by adding user fairness terms. We urge designers of fair recommendation systems in practice to develop such evaluations to measure such individual-level effects, and for researchers to further characterize the potential implications of such algorithms. Our code is available at the following repository: https://github.com/vschiniah/ArXiv_Recommendation_Research.

## 2 Formal Model

Our setup is characterized by user and item (estimated) utilities, recommendation optimization with user and item fairness desiderata, and evaluation of the effects of such desiderata. (As discussed below, some of these modeling choices are made for concreteness, and our results extend beyond these specific choices).

**Users, items, and utilities.** There is a finite population of $m$ users and $n$ items. Let $w_{ij} > 0$ be the utility of recommending item $j$ to user $i$; this could represent a click-through rate or purchase probability. We suppose that user-item utilities are symmetric: each of the user $i$ and item $j$ receives utility $w_{ij}$ from being recommended item $j$.

Let $\Delta_{n-1}$ denote the simplex in $\mathbb{R}^n$. The platform's task is to choose a recommendation policy $\rho \in \Delta_{n-1}^m$. For each user $i$, the platform will recommend one item to user $i$ selected randomly according to the distribution $\rho_i$. Given a recommendation policy $\rho$, user $i$'s expected utility from using the platform is $\sum_j \rho_{ij} w_{ij}$; item $j$'s expected utility is $\sum_i \rho_{ij} w_{ij}$, where $\sum_j \rho_{ij} = 1$ for each user $i$.

**Fairness desiderata.** We suppose that the platform uses as a benchmark, for each user or item, the *best* thing that it could do for that agent if it ignored the utilities of others. Thus, given a recommendation policy $\rho$, let $U_i(\rho, w)$ be user $i$'s utility from $\rho$, normalized by the utility $\max_j w_{ij}$ they would receive from being recommended their best match. Let $I_j(\rho, w)$ be item $j$'s utility from $\rho$, normalized by the utility $j$ receives if it is recommended to *every* user, $\sum_i w_{ij}$. The normalized utilities are:

$$U_i(\rho, w) = \frac{\sum_j \rho_{ij} w_{ij}}{\max_j w_{ij}}, \quad I_j(\rho, w) = \frac{\sum_i \rho_{ij} w_{ij}}{\sum_i w_{ij}}.$$

The normalizations capture that recommendations should not be affected by scaling utilities, or be distorted by a user who is not satisfied with any item or an item that is generally undesirable to users.

Given a recommendation policy $\rho$, user fairness is quantified as the *minimum* normalized user utility $U_{\min}(\rho, w) = \min_i U_i(\rho, w)$, and item fairness analogously as $I_{\min}(\rho, w) = \min_j I_j(\rho, w)$.

**Multi-objective recommendation optimization.** We suppose that the platform seeks to satisfy its fairness desiderata as follows. At the extremes, the platform could choose a maximally fair solution for one side, ignoring the other. Denote the optimal user fair utility as $U_{\min}^*(w) := \max_\rho U_{\min}(\rho, w)$ (achieved by giving each user their favorite item deterministically), and the optimal item fair utility as $I_{\min}^*(w) := \max_\rho I_{\min}(\rho, w)$.

Finally, we cast the two-sided fair optimization – for the optimal $\gamma$-constrained user fair solution – as

$$U_{\min}^*(\gamma, w) = \max_\rho \quad U_{\min}(\rho, w) \tag{1}$$
$$\text{subject to} \quad I_{\min}(\rho, w) \geq \gamma I_{\min}^*(w),$$

i.e., we maximize the minimum normalized user utility, subject to the minimum normalized item utility being at least a fraction $\gamma$ of the optimal item fair solution.

**Research questions: price of fairness and misestimation.** We can now define the price of item fairness on user fairness $\pi_{U|I}^F$ ("price of fairness") as the decrease in user fairness with maximal item fairness constraints:

$$\pi_{U|I}^F(w) := \frac{U_{\min}^*(w) - U_{\min}^*(\gamma = 1, w)}{U_{\min}^*(w)}.$$

We ask how $\pi_{U|I}^F$ changes with the utility matrix $w$. We note that while this question (in terms of solutions to Problem (1)) can be simply stated, as we detail in Section 3, finding a closed form expression for $U_{\min}^*(1, w)$ – the optimal minimum normalized user utility given item fairness constraints – in terms of $w$ is theoretically challenging.

Similarly, we investigate the price of misestimation. Let $\hat{w}_{ij}$ denote the platform's *estimate* of the utility of recommending $i$ to $j$; let $\hat{\rho}(\gamma)$ be a policy that solves the optimization problem above with

misestimated utilities, that is, $\hat{\rho}(\gamma)$ attains $U_{\min}^*(\gamma, \hat{w})$.[1] Then we define the price of misestimation on user utility ("price of misestimation") $\pi_U^M$ as

$$\pi_U^M(\gamma, w, \hat{w}) := \frac{U_{\min}^*(\gamma, w) - U_{\min}(\hat{\rho}(\gamma), w)}{U_{\min}^*(\gamma, w)},$$

that is, the decrease in true user fairness due to optimizing with estimated utilities. In particular, in Section 5 we will examine whether item fairness exacerbates the price of misestimation by comparing $\pi_U^M(0, w, \hat{w})$ and $\pi_U^M(1, w, \hat{w})$, particularly in the case of cold start users.

**Discussion.** We note several modeling choices. First, we assume that utility $w_{ij}$ is shared – both the item $j$ and user $i$ benefit equally from a successful recommendation. This choice captures, e.g., purchase or click-through rates. It also helps us isolate effects due to fairness constraints and effects across items and users, as opposed to misaligned utilities. We expect the price of fairness to increase – and potentially be arbitrarily high – with such misaligned utilities. We further justify and relax the shared utility assumption in Appendix A. Second, we quantify fairness through the minimum normalized utility. Normalization is standard in related algorithmic work, such as [47], and avoids solutions in which an item that provides utility $\epsilon$ to every user needs to be recommended to every user to equalize utilities. We use *individual egalitarian* fairness (minimum utility over individuals) instead of group fairness to capture settings in which group identity is not available, and systems in which individual-level disparities may be widespread; egalitarian fairness is widespread in algorithmic fairness [2, 17, 43, 45]. In Appendix A we show that our empirical findings extend to other measures of individual fairness. Finally, in the user-item fairness tradeoff problem (Problem 1) we used an item fairness constraint rather than adding an item fairness term to the objective; these two approaches can be thought of conceptually as dual to one another, and so we expect similar properties to hold in either formulation.

# 3 Theoretical framework

To determine how the price of fairness $\pi_{U|I}^F$ depends on utility matrix $w$, we need to compute $U_{\min}^*(1, w)$ and $U_{\min}^*(w)$. It turns out that $U_{\min}^*(w)$ is easy to describe: without item fairness constraints, the optimal recommendation policy deterministically recommends each user their most preferred item. Each user attains the maximum possible normalized utility of 1, and $U_{\min}^*(w) = 1$.

However, with an item fairness constraint, we can no longer select each user's recommendation policies independently. Thus characterizing $U_{\min}^*(1, w)$ is much more complicated. Recall that $U_{\min}^*(1, w)$ is the minimum normalized user utility of the optimal user-fair recommendation subject to maximal item fairness constraints. Plugging into Problem (1) and expanding the definitions, we have

$$U_{\min}^*(1, w) = \max_{\rho \in \Delta_{n-1}^m} \quad \min_i \frac{\sum_j w_{ij} \rho_{ij}}{\max_j w_{ij}} \tag{2}$$

$$\text{subject to} \quad \rho \in \arg \max_{\phi \in \Delta_{n-1}^m} \min_j \frac{\sum_i w_{ij} \phi_{ij}}{\sum_i w_{ij}}.$$

Thus we need to find closed-form solutions in terms of the utilities $w$ to a non-linear concave program, in which both the objective and the constraint depend on $w$, and indeed even determining the constraint requires solving another non-linear concave program.

In Proposition 1, we develop a framework to solve this problem, which we later apply to show our main results Theorems 3 and 4.

**Proposition 1.** *Suppose that for a set of recommendation policies $\mathcal{S} \subseteq \Delta_{n-1}^m$,*

*(i) $\mathcal{S}$ can be described by a finite set of linear constraints*

*(ii) There exists an optimal solution $\rho^*$ to Problem (2) such that $\rho^* \in \mathcal{S}$*

*(iii) $\rho^*$ is the unique feasible solution to Problem (2) in $\mathcal{S}$*

---

[1] Multiple policies $\hat{\rho}$ may optimize $U_{\min}^*(\gamma, \hat{w})$, and different policies $\hat{\rho}$ may have different values of $U_{\min}(\hat{\rho}, w)$. In our experimental results, we use the policy $\hat{\rho}$ found by the convex optimization solver.

*Then, finding an optimal solution $\rho^*$ to Problem (2) can be reduced to solving a linear program $\mathcal{L}$.*

With Proposition 1, the key technical challenges become: (a) given utility matrix $w$, finding $\mathcal{S}$ satisfying the above conditions; (b) given $\mathcal{S}$ and $w$, finding a closed-form expression for $\rho^*$; (c) given a closed form for $\rho^*$ in terms of $w$, reasoning about the properties of item fair solution $U_{\min}^*(1, w)$.

To do this, for our main results, we construct $\mathcal{S}$ as a set of recommendations with a particular *symmetric* structure. Suppose users come in $K$ types, where a user of type $k$ shares a utility vector. Then, consider $\mathcal{S}_{\mathrm{symm}} = \{\rho : \rho_i = \rho_{i'} \text{ if } w_i = w_{i'}\} \subseteq \Delta_{n-1}^m$, the set of policies where users of the same type are given the same recommendation probabilities. Note that, in the extreme case, each user is their own type. Furthermore, let $\rho_k$ denote the recommendation policy for type $k$, for $\rho \in \mathcal{S}_{\mathrm{symm}}$.

**Proposition 2.** *$\mathcal{S}_{symm}$ satisfies conditions (i) and (ii) in Proposition 1. Furthermore, solutions $\rho \in \mathcal{S}_{symm}$ to Problem (2) have a sparse structure:*

- *If $\rho$ is a basic feasible solution to the linear program $\mathcal{L}$ in Proposition 1, then there are at most $n + K - 1$ type-item pairs $(k, j)$ such that $\rho_{kj} > 0$ (out of $nK$ possible pairs).*

- *If $\rho$ is also optimal, then there are at most $K - 1$ items that are ever recommended to more than one type of user, i.e., where $\rho_{kj}, \rho_{k'j} > 0$ for $k \neq k'$.*

For our main results, we will leverage the sparsity structure in Proposition 2 to show, with further restrictions on utility matrix $w$, that $\mathcal{S}_{\mathrm{symm}}$ also satisfies $(iii)$. We will then derive closed-form expressions for that solution, and show how it changes as a function of $w$.

The sparsity structure in Proposition 2 is also interesting in its own right. Without item fairness constraints, solutions will be highly sparse: each user will be recommended their most preferred item deterministically; each other item will have a recommendation probability of zero. Once we add item fairness constraints, solutions do not necessarily remain sparse. However, Proposition 2 shows that sparse solutions still arise under item fairness constraints in settings with symmetric solutions.

## 4 User preference diversity and fairness tradeoffs

We now use the theoretical framework developed in the above section to understand the effect of the structure of user utilities on the price of fairness. In particular, we identify "free fairness," i.e., the price of fairness is low, when preferences are sufficiently diverse.

**Example 1.** For intuition as to why user diversity affects the price of fairness, consider the following example. Suppose we have $n = 2$ items and $m$ users; half the users have utility $\epsilon$ for the first item and $1 - \epsilon$ for the second item, and the other half has utility $1 - \epsilon$ for the first item and $\epsilon$ for the second. Then, the recommender can simply give each user their favorite item (the user optimal solution), and this solution simultaneously maximizes user and item fairness as well as total user and item utility.

On the other hand, suppose all users have the same preferences: each user has utility $1 - \epsilon$ for the first item and utility $\epsilon$ for the second item, for $\epsilon > 0$ that is small. Then, *any* recommendation probability given to the second item comes at a cost to users who receive that item instead of the first item; however, (normalized) item fairness would require that the second item receives $\epsilon$ as much utility as the first item. Below, we show that this results in a tradeoff even between linear *normalized* user and item utilities, where we account for the fact that the second item is on average less preferred by users.

Since all users have the same preferences for items, using Proposition 2 it is sufficient to consider them receiving the same recommendation probabilities $\rho_1$ for the first item and $\rho_2 = 1 - \rho_1$ for the second. Bounding minimum item utility $I_{\min}$ by the utility of the second item, we have

$$I_{\min} \leq I_2(\rho) = \frac{\sum_i \rho_2 \epsilon}{m\epsilon} = \rho_2 \triangleq 1 - \rho_1.$$

For a given recommendation probability $\rho_1$, the minimum normalized user utility is

$$U_{\min} = (1 - \epsilon)\rho_1 + \epsilon\rho_2 = \epsilon + (1 - 2\epsilon)\rho_1 \leq \rho_1 + \epsilon.$$

Rearranging, we get that

$$U_{\min} - \epsilon \leq \rho_1 \leq 1 - I_{\min} \implies U_{\min} + I_{\min} \leq 1 + \epsilon.$$

Thus the minimum normalized user utility and the minimum normalized item utility essentially follow a negative *linear* relationship – guaranteeing the second item even $\epsilon$ as much utility as the first item results in a linear cost to users. □

We now formalize and generalize this example, when the level of heterogeneity in the population can be captured by a single parameter. Consider the following utility matrix structure. Let $v_1 > v_2 > ... > v_n > 0$. Suppose that there are two user types: a user $i$ either has utility $w_{ij} = v_j$ or $w_{ij} = v_{n-j+1}$, and say that user $i$ is of type 1 or 2 respectively. In words, the two types have opposite preferences, but the preferences can otherwise be generic and be for any number of items. Now, the direct solution by symmetry for the example no longer directly holds – the items in the middle, not necessarily preferred by either user type, may be binding in terms of item fairness constraints.

Let $\alpha$ be the proportion of type 1 users in the population, out of a fixed population of $m$ users. Parameter $\alpha$ thus controls the population heterogeneity; if $\alpha$ is near 0 or 1, the population is highly homogeneous, dominated by users of the same type. If $\alpha = \frac{1}{2}$, the population is split evenly between the two types and is highly heterogeneous. Since we parametrize $w$ by $\alpha$, we may write the price of fairness as,

$$\pi_{U|I}^F(\alpha) := \frac{U_{\min}^*(\alpha) - U_{\min}^*(\gamma = 1, \alpha)}{U_{\min}^*(\alpha)}.$$

Given this structure, Theorem 3 states that the price of fairness $\pi_{U|I}^F(\alpha)$ increases in the homogeneity of the users – heterogeneous user populations are less affected by incorporating item fairness constraints.

**Theorem 3.** $\pi_{U|I}^F(\alpha)$ *is decreasing in $\alpha$ for $0 < \alpha \leq 1/2$, and increasing in $\alpha$ for $1/2 \leq \alpha < 1$.*

*Proof sketch for Theorem 3.* We show that when in a population with two opposing types as described above, the sparsity condition in Proposition 2 yields a unique solution, for which we can find a closed form and express in terms of $\alpha$. We then evaluate $U_{\min}(\rho, \alpha)$ at this solution to find $U_{\min}^*(1, \alpha)$, and show that this is indeed increasing. The full proof is in Appendix D. □

## 5 Uncertainty and fairness tradeoffs

A basic fact – often ignored in fair recommendation – is that recommendations are made with (mis)estimated utilities. Platforms do not have full knowledge of user preferences, especially those new to the platform. Of course, recommendations under misestimated utilities may be poor; here, we show that adding item fairness constraints may *worsen* the cost of this misestimation even further.

Intuitively – for a new user for whom the platform has no data – the platform would estimate the user's preferences as the average of preferences of existing users (e.g., in a Bayesian fashion). Thus, without item fairness considerations, it would show the user generally popular items. However, the new user's preferences are generally estimated as "weaker" than the preferences of others for any given item (since it averages preferences of users who may either like or dislike any given item). Thus, with fairness constraints, the optimization is incentivized to show the user otherwise unpopular items, since all the user's estimated preferences are weaker. Liu and Burke [31] for example develop an algorithm for item fairness where users with weaker preferences are explicitly leveraged in this way.

For a given item fairness level $\gamma$, true utility matrix $w$, and estimated utility matrix $\hat{w}$, let $\hat{\rho}(\gamma)$ be a recommendation policy that solves the recommendation problem (Problem 1) with respect to the misestimated utilities, that is, $\hat{\rho}$ solves $U_{\min}^*(\gamma, \hat{w})$. Recall that we define the price of misestimation

$$\pi_U^M(\gamma, w, \hat{w}) = \frac{U_{\min}^*(\gamma, w) - U_{\min}(\hat{\rho}(\gamma), w)}{U_{\min}^*(\gamma, w)},$$

which represents the relative decrease in minimum normalized user utility as a result of misestimation. Item fairness *worsens* the price of misestimation if $\pi_U^M(\gamma = 1, w, \hat{w}) > \pi_U^M(\gamma = 0, w, \hat{w})$.

We now formalize the above argument, building on the analysis in the previous section. As in Section 4, suppose that there are 2 types of users, with opposing preferences (i.e., with values $v_1, v_2..., v_n$ and $v_n, v_{n-1}, ..., v_1$, respectively) – and the platform has correctly estimated these preferences. However, now, these two types only make up a proportion $\beta$ of the population each.

Now, we suppose that there is a fraction $1 - \beta$ of the user population who are "new" users. We assume that these users are drawn from the same distribution as the remaining users, but the platform does not know their preferences. It thus constructs a prior by averaging over the known users' preferences – (mis)estimating the users' utility for each item $j$ as $\frac{v_j + v_{n-j+1}}{2}$.

**Theorem 4.** *If $\beta > \frac{1}{n}$ and $w$ and $\hat{w}$ are as described above, then fairness constraints can arbitrarily worsen the price of misestimation.*

- *The price of misestimation without fairness constraints is low: for all $\{v_j\}$, there is a recommendation policy $\hat{\rho}$ that solves the misestimated problem $U^*_{\min}(0, \hat{w})$ so that*

$$\pi^M_U(0, w, \hat{w}) \leq \frac{1}{2}.$$

- *The price of misestimation with fairness constraints can be arbitrarily large: $\forall \epsilon$, there exists $\{v_j\}$ and a recommendation policy $\hat{\rho}$ that solves the problem $U^*_{\min}(1, \hat{w})$ such that*

$$\pi^M_U(1, w, \hat{w}) > 1 - \epsilon.$$

*Proof sketch.* The main task is to find the price of misestimation with fairness constraints, which requires computing $U^*_{\min}(1, w)$ when users' values are correctly estimated, and computing $U^*_{\min}(1, \hat{w})$ when users' values are incorrectly estimated. To find $U^*_{\min}(1, w)$, note that $w$ is a population with two opposing types of users, so we may leverage the insights of Theorem 3. To find $U^*_{\min}(1, \hat{w})$ we again use the framework in Section 3, showing that in the setting of this theorem, we can find an optimal policy $\rho^*$ for $U^*_{\min}(1, \hat{w})$ in a set $\mathcal{S}' \subseteq \mathcal{S}_{\text{symm}}$ of policies with an additional *column*-symmetry property. We use an analogue of the sparsity result in Proposition 2 to show that there is a unique feasible solution $\hat{\rho} \in \mathcal{S}'$ and obtain a closed form expression for $\rho^*$, and find an upper bound for $U_{\min}(\hat{\rho}, \hat{w}) = U^*_{\min}(1, \hat{w})$. In fact, we show that as long as $\beta > \frac{1}{n}$, under $\hat{\rho}$ the mis-estimated users will *never* be recommended their most preferred items. The full proof is in Appendix E. $\qquad \square$

The idea follows the above intuition: without item fairness constraints and assuming cold start users follow the same distribution as existing users, the platform's price of misestimation is low because the platform treats cold start users as the average of the existing population. Fairness constraints, however, can make this cost arbitrarily high, as *in expectation* cold start users *relatively* enjoy items other users do not. Such effects suggest that a more careful treatment of uncertainty and fairness together is necessary for recommendation algorithms.

## 6 Empirical findings: arXiv recommendation engine

We prototype a recommender for preprints on arXiv, to illustrate our conceptual findings. We consider the cold start setting for items (papers), when they are newly uploaded to arXiv and so only have metadata and paper text but no associated interaction or citation data. For users (readers), we use as data the papers that *they* have shared on arXiv to estimate their preferences.

**Empirical setup.** We use data from arXiv and Semantic Scholar [1, 25]. As training for user preferences, we consider 139,308 CS papers by 178,260 distinct authors before 2020; as the items to be recommended, we consider the 14,307 papers uploaded to arXiv in 2020. We apply two natural language processing-based models – TF-IDF [28] and the sentence transformer model SPECTER [13] – to textual features such as the paper's abstract (for both items and the user's historical papers) to generate embeddings for all papers in the training set. We use these embeddings to compute similarity scores (utility matrices) for users and items. To compute the similarity score (utility) between a user (an author of at least one paper before 2020) and an item (a paper uploaded in 2020), we compute the cosine similarity between the embedding of each of the user's pre-2020 papers and the item's embedding. We then use either the *mean* or the *max* similarity amongst the pre-2020 papers and the item; the *max* similarity score may more effectively capture a user's diverse interests [20, 41]. We then generate recommendations for each user, at various levels of user and item fairness constraints.

To validate our recommendation approach, we use citation data from Semantic Scholar [25] to determine for each user and each paper published in 2020 whether the user cites that paper in their post-2020 work. We then examine how well the user-item similarity score generated by our recommendation engine predicts the presence of a citation. The recommendations effectively predict whether a user cites a paper with a high score after 2020. In Table 1 we show the results of a logistic regression between each similarity score and the presence of a citation, where the coefficient on the score is large and statistically significant for each model; the predictive power of our models is

| Model | Coefficient | Std. Err | z-value | Adjusted $R^2$ |
|---|---|---|---|---|
| Max score, TF-IDF | 12.4100 | 0.058 | 212.178 | 0.08915 |
| Mean score, TF-IDF | 20.2122 | 0.131 | 154.835 | 0.04616 |
| Max score, Sentence transformer | 18.4557 | 0.250 | 73.695 | 0.1347 |
| Mean score, Sentence transformer | 16.2482 | 0.246 | 66.148 | 0.09085 |

Table 1: Logistic regression results for predicting whether user $i$ cites paper $j$ from the similarity score $w_{ij}$ for each model.

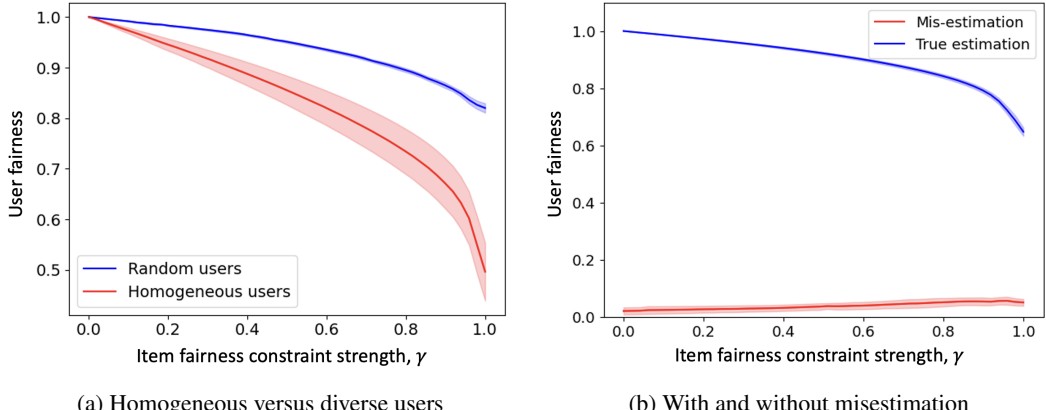

(a) Homogeneous versus diverse users          (b) With and without misestimation

Figure 1: Empirical (using our arXiv recommender) tradeoff between the minimum user (Y axis) and item (X axis) utility. Recall $\gamma$ is the fraction of the best possible minimum normalized item utility $I_{\min}^*$ guaranteed. (a) Illustrating Theorem 3 empirically – homogeneous populations have a higher price of fairness. Empirically, however, the price of fairness is small except with strict item fairness constraints $\gamma \to 1$. (b) For a set of users, holding other users fixed, the cost to the worst-off user of misestimating preferences, at varying $\gamma$. Empirically, the cost of misestimation is already so high that it is not worsened with item fairness constraints, as in the worst case analysis of Theorem 4.

reasonable for a sparse, high variance event such as citations. We generally find that the *max* score models are more predictive of future citations. Appendix B includes details and evaluations; our computational experiments in the main text use the max score, TF-IDF model.

## 6.1 Empirical results.

We examine the tradeoffs between item and user fairness and the effect of misestimation on this tradeoff empirically. We use the similarity scores generated by the recommendation engine described above – in particular, the max score, TF-IDF model – as the utility values $w_{ij}$. We consider a pool of 14,307 papers in the computer science category posted to arXiv in 2020, and a pool of 20,512 authors who posted papers in the computer science category to arXiv both in 2020 and prior to 2020 (we use the papers prior to 2020 to compute the similarity scores as described above). We then subsample recommendation settings from this pool; we give further details about the sampling process below. For a given value of $\gamma$, to compute $U_{\min}^*(\gamma)$ we use the cvxpy implementation of the convex optimization algorithm SCS [38].

**User-item tradeoffs as function of user diversity.**   Figure 3a shows the tradeoff between user fairness and item fairness in a random population and in a population of homogeneous users. To generate a tradeoff curve for the heterogeneous population, we sampled 200 random papers and 500 random authors to form $w$, and computed $U_{\min}^*(\gamma, w)$ for 50 values of $\gamma$ between 0 and 1. To generate tradeoff curves from homogeneous user populations, we clustered all 20,512 users into 10 clusters using the $k$-means algorithm. For a single curve, to form $w$ we sampled 200 random papers and 500 random authors from one random cluster. We run 10 experiments and plot the mean of $U_{\min}^*(\gamma, w)$ at each value of $\gamma$ across the 10 curves as well as two std. error bars for the mean.

Figure 3a demonstrates that in real data, for moderate item fairness guarantees ($0 \leq \gamma \leq 0.9$), on average there is a fairly low cost to user fairness, but as we approach optimal item fairness ($\gamma \to 1$), the tradeoff becomes steep. Furthermore, Figure 3a shows that item fairness tends to impose a higher cost to user fairness in more homogeneous populations, as in Theorem 3.

**Price of misestimation.** Figure 3b shows how user fairness is affected by item fairness guarantees in the presence of misestimation. For sets of 200 random papers and 500 random authors, we select a set 10% of these users at random and treat them as if we did not have any data for them, estimating these users' utility for item $j$ as the average utility for item $j$ for the other 90% of the users ($\hat{w}$ in the notation of Theorem 4). For 50 values of $\gamma$ between 0 and 1, we compute $U_{\min}^*(1, \hat{w})$ for the misestimated utility matrix. We also compute, counterfactually, this quantity if the utility matrix had been correctly estimated, $U_{\min}^*(1, w)$. We plot the true minimum normalized user utility under the recommendation policies that attain $U_{\min}^*(1, w)$ and $U_{\min}^*(1, \hat{w})$. We run 10 experiments and plot the mean value of the minimum normalized user utility and two std. error bars.

In Figure 3b, near $\gamma = 0$, the price of misestimation – the gap between the two curves – is higher than the cost of misestimation near $\gamma = 1$. This results because the item fairness constraint barely changes the (already low) utility of these users when their preferences are misestimated, but substantially changes their utility when their preferences are correctly estimated. While theoretically, item fairness constraints can arbitrarily increase the cost of misestimation, in practice, on average they do not affect this cost – the cost of misestimation without item fairness is already high.

# 7 Related work

There is a large literature on (item) fair recommendation and ranking [3, 35, 48, 50, 51] and, more recently, on multi-sided user-item fair recommendation [5, 10–12, 16, 39, 47]. This literature is primarily *algorithmic:* for a given formulation of user and item utility (and other desiderata), how do we devise an efficient algorithm for multiple objectives or constraints? In contrast, our goal is primarily *conceptual*, to aid algorithm designers in choosing when to use such algorithms: for example, by explaining when we might expect the tradeoff to be especially sharp, and to understand the cost to cold start users in particular. For example, we theoretically analyze recommendations from a constrained optimization-based approach akin to that in Basu et al. [5], in terms of the implications of such an approach on recommendations.

Several papers observe related phenomena to the ones we study, especially empirically. Wang and Joachims [47] develop an optimization algorithm to be fair to users (in terms of group fairness to demographic groups) and items (similar to our normalized utility metric). Theoretically, they show that there is a tradeoff between user and item utility metrics, and further empirically show how fairness interacts with the diversity of items shown to each user. While their focus is also primarily algorithmic, they do show that there is a fundamental tension between item and user fairness. In concurrent work, Kleinberg and Meister [26] also theoretically demonstrate and characterize this tension. They focus on the cost to individual users caused by imposing maximal item fairness constraints – similar to how we define price of fairness – and determine the relationship between a cost level and the proportion of users in the worst-case recommendation setting who experience that cost. In contrast, we examine the cost to the single worst-off user as a function of properties of the recommendation setting such as user diversity and recommender mis-estimation. Rahmani et al. [42] demonstrate a similar tension between item and user fairness in empirical data. Liu and Burke [31] do not examine user fairness, but observe that one can mitigate the cost of item fairness in multi-sided recommendations by recommending to users with weaker preferences for items that may otherwise be less preferred. Subsequently, Farastu et al. [16] examine which users bear the cost of item fairness, pointing to Liu and Burke [31] to argue that the cost to users of item fairness constraints disproportionately falls on users with flexible preferences, creating an incentive for users to misrepresent their preferences as more rigid than reality. We build on these arguments by theoretically analyzing the cost of item fairness on (individual) users, especially the users most affected, as a function of the estimated user preference matrix.

In focusing on conceptual phenomena, our work is related to work analyzing the *price of fairness* and efficiency-equity tradeoffs in various settings beyond recommendations. Bertsimas et al. [8] first defined the *price of fairness* as the normalized decrease in the utility of an algorithmic outcome after adding fairness considerations, and develop general bounds on the price of fairness in an array of

optimization scenarios. This concept has been subsequently applied in a variety of domains, including auction theory [24], fair division and resource allocation [6, 32, 46], and computer networking [49]. Barre et al. [3] apply a similar concept in assortment optimization, examining the cost of item visibility constraints on the revenue of a platform. Most similarly, Chen et al. [12] define the price of fairness in the setting of multi-sided fairness as the cost on platform revenue of including both fairness constraints; in contrast we define the price of fairness as the cost to user fairness of imposing item fairness constraints in order to capture the interplay between user and item fairness. The authors then show that this price of fairness on the revenue depends on *objective misalignment* – the difference in fairness between the item/user utility required by the constraints, and the item/user utility in a revenue-optimal solution. We study how the price of item fairness on user fairness depends on *user preference diversity* – the agreement between users' utilities. These concepts are related: user preference diversity may cause objective alignment. Moreover, they also consider the problem of unknown preferences: they examine how to algorithmically impose fairness constraints when preferences are unknown, while we address the question of whether fairness constraints disproportionately harm users with unknown preferences. Finally, our result that diverse population preferences can mitigate the price of fairness resembles the result of Bastani et al. [4] that greedy contextual bandits can perform well without exploration if there is sufficient contextual diversity.

Finally, there is a large literature on other tradeoffs in recommendations, rankings, and ratings: engagement versus value, diversity, strategic behavior, uncertainty, and over-time dynamics [9, 14, 15, 18–23, 27, 29, 30, 33, 34, 36, 41]. In the context of set recommendations when users consume their favorite item out of the multiple recommended, e.g., Peng et al. [41] show that there is a minimal utility-diversity tradeoff, and Besbes et al. [9] show that there is a minimal exploration-exploitation tradeoff.

# 8 Discussion

We investigate the relationship between user fairness and item fairness in recommendation settings. We develop (a) a theoretical framework to enable us to solve for the price of fairness for many population settings, and (b) a recommendation engine using real data to allow us to investigate user-item fairness tradeoffs in practice. Our work informs the design of fair recommendation systems: (1) it emphasizes the benefits of a diverse user population, and suggests that item fairness constraints should not be imposed on *sub-markets* (sub-groups), but instead on the entire population together. (2) It cautions designers to be especially mindful of effects on individual users (especially cold start users), who may receive disproportionately poor recommendations with item fairness constraints, even with a user fairness objective. Our empirical analysis supports our theoretical analysis—the userbase diversity affects the severity of user-side effects of imposing item fairness; however, the price of misestimating user utility is already high without item fairness constraints, and so imposing such constraints does not have additional effects. Such results speak to the importance of instance-specific analyses, cf. [40]: one cannot make general statements about the specific effects of item-fairness constraints on users (or vice versa) outside of a specific context, though we identify two relevant phenomena (user diversity and misestimation) that modulate these effects.

**Limitations.** Our theoretical analysis explores these fairness tradeoffs in a fairly restricted setting. First, we assume that users are only recommended a single item; future work should investigate how the price of fairness changes as the number of recommended items increases. We do not expect our theoretical framework to easily extend to other definitions of fairness; however, we extend our computational arXiv experiments to other definitions of fairness in Appendix A. These extended experiments also show that diverse user preferences reduce fairness tradeoffs; an interesting direction for future work is to theoretically characterize user-item fairness tradeoffs under other definitions of fairness. Furthermore, in practice platforms are unwilling to maximize the worst-off user's item or user fairness at the expense of the entire platform's utility; the problem is really one of balancing user and item fairness with overall platform performance. Algorithms to optimize these multi-sided problems are explored in other work [12], but it would be interesting to develop qualitative observations about user-item fairness tradeoffs in the presence of a total utility constraint. Finally, we show our Theorems 3 and 4 in a limited context with only two or three types of users. However, the theoretical framework developed in Section 3 is significantly more general. It is would be interesting to apply this framework to other population structures such as when users do not have perfectly opposite preferences and where there are more than three groups of users.

## Acknowledgments and Disclosure of Funding

SG is supported by a fellowship from the Cornell University Department of Computer Science and an NSERC PGS-D fellowship [587665]. NG is supported by NSF CAREER IIS-2339427, and Cornell Tech Urban Tech Hub, Meta, and Amazon research awards. The authors would like to thank Sidhika Balachandar, Erica Chiang, Evan Dong, Omar El Housni, Meena Jagadeesan, Jon Kleinberg, Kevin Leyton-Brown, Michela Meister, Chidozie Onyeze, Kenny Peng, Emma Pierson, and Manish Raghavan for valuable conversations and insights, as well as the NeurIPS 2024 reviewers for helpful feedback.

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

# A Extensions

In the experiments below, we subsample recommendation settings in the same way as described in Section 6 but with 50 papers and 100 authors.

## A.1 Alternative definitions of fairness

**Nash Welfare** The first alternative definition of fairness we consider is Nash welfare [37],

$$U_{\mathrm{NW}}(\rho) = \sum_i \log U_i(\rho), \quad I_{\mathrm{NW}}(\rho) = \sum_j \log I_j(\rho).$$

This measure of fairness is more holistic than egalitarian fairness as it accounts for the utilities of all users (items). In Figure 2, we show the results of repeating the experiment in Figure 1 with Nash Welfare fairness. We see that the trade-off between user and item fairness is steeper for homogeneous populations of users than for uniformly random populations. While the curves appear more concave than before, it is important to notice that we replaced $\gamma$ with $1/\gamma$ in the item fairness constraint of the optimization problem due to the Nash welfare being negative (as detailed in Figure 2), so that $\gamma$ – while still capturing constraint strength – has a slightly different interpretation than previously.

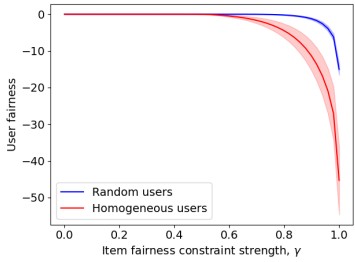
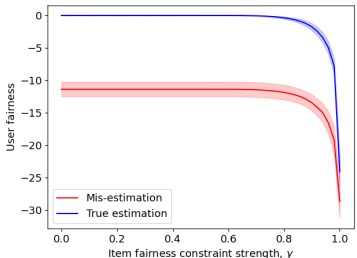

(a) Homogeneous versus diverse users      (b) With and without misestimation

Figure 2: We repeat the experiment of Figure 1 from the original paper but replace max-min fairness with Nash welfare fairness. That is, in the objective we replace $U_{\min}$ with the user Nash welfare $U_{\mathrm{NW}}$. We must be more careful with the item fairness constraint: we know that the normalized utilities satisfy $0 \leq U_i, I_j \leq 1$, so $U_{\mathrm{NW}}, I_{\mathrm{NW}} < 0$. This means that in Problem 1 we must replace the item fairness constraint $I_{\min}(\rho) \geq \gamma I_{\min}^*$ with the constraint $I_{\mathrm{NW}}(\rho) \geq (1/\gamma) I_{\mathrm{NW}}^*$. When $\gamma = 0$, this corresponds to $I_{\mathrm{NW}}(\rho) \geq -\infty$; when $\gamma = 1$, this corresponds to $I_{\min}(\rho) \geq I_{\min}^*$. Thus as before, when $\gamma = 0$ there is effectively no item fairness constraint, and $\gamma = 1$ constrains item fairness to be maximal.

**Sum of $k$-min** The second alternative definition of fairness we consider is the sum of the $k$-minimum user or item utilities, which is a generalization of egalitarian fairness ($k = 1$) that measures the utility of a size-$k$ *set* of worst-off entities,

$$U_{k-\min}(\rho) = \min_{i_1 \neq \ldots \neq i_k} \sum_{\ell=1}^{k} U_{i_\ell}(\rho), \quad I_{k-\min}(\rho) = \min_{j_1 \neq \ldots \neq j_k} \sum_{\ell=1}^{k} I_{j_\ell}(\rho).$$

In Figure 3, we show the results of repeating the experiment in Figure 1 with max-sum-$k$-min fairness. We again observe that the trade-off between user and item fairness is steeper for homogeneous populations of users than for uniformly random populations. We also do not see an increase in the price of mis-estimation when item constraints are added – again, the price of mis-estimation is very high.

## A.2 Alternative item utility models

In our theoretical and empirical results, we use a symmetric utility model, where an item's utility for being recommended to a user is the same as the user's utility for the recommendation. Formally, in general item $j$ has utility $w_{ij}^I$ for being recommended to user $i$, while user $i$ may have a different

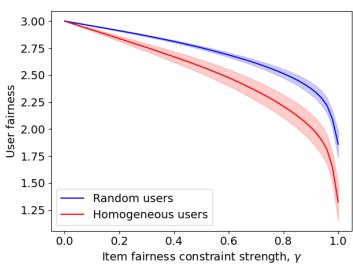
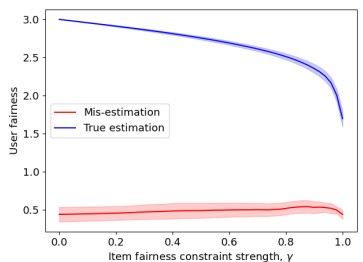

(a) Homogeneous versus diverse users        (b) With and without misestimation

Figure 3: We repeat the experiment of Figure 1 from the original paper but replace max-min fairness with max-sum-$k$-min fairness for $k = 3$. In the optimization in Problem 1, we replace $U_{\min}$ and $I_{\min}$ with $U_{k-\min}$ and $I_{k-\min}$ respectively.

utility $w_{ij}^{U}$ for that recommendation. In our theoretical and empirical results above, we took $w_{ij} = w_{ij}^{I} = w_{ij}^{U}$. Below, we discuss the rationale for this model, and show that this assumption may be relaxed in our theoretical and empirical results.

The symmetric utility model we use – which corresponds to the "market share" item utility model in Chen et al. [12] – is motivated by platforms in which both users and items derive utility from a *successful* recommendation. This is reasonable in settings in which not only does a user want to be recommended relevant items, but an item's producer wants it to be recommended to users for which the item is especially relevant. Concretely, in the case of readers receiving recommendations for academic pre-prints, the readers prefer papers that they will engage with, and authors prefer readers that will engage with their work. in an online marketplace producers want customers who will purchase their product to be shown the item; in a social media setting, content creators prefer their content to appear to users who will appreciate it and thus engage with future content. Our symmetric utility model captures this basic structure behind producer preferences in many cases.

Note that since users can only receive a limited number of recommendations, our assumption does not eliminate the tension between user and item utility: an individual user wants recommendations that maximize her total utility across all items, while an individual item wants the platform to produce recommendations that maximize its total utility across all users. A concrete example of this is that under this model a low-quality item will want to be recommended to the user most likely to click on it, but that user won't want to be recommended the low-quality item. This assumption does, however, imply that the *platform-wide* item utility and *platform-wide* user utility are equivalent – for recommendations $\rho_{ij}$, these are both $\sum_{j} \sum_{i} w_{ij} \rho_{ij}$.[2]

One generalization of the symmetric utility model is where each user and item receives utility *proportional* to some shared recommendation quality $w_{ij}$. Formally, each user $i$ receives a utility of $a_i w_{ij}$ and each item $j$ receives a utility of $b_j w_{ij}$ from recommending $j$ to $i$, for $a_i, b_j > 0$. For example, different values of $a_i$ could capture different levels of baseline interest in items among users. Our theoretical results still hold in this more general setting, since these coefficients will cancel out when we normalize the utilities.

Another common item utility model is *exposure* [11, 39], where items receive utility only from being recommended, and are ambivalent to *which* user it is shown to. Formally, this corresponds to taking $w_{ij}^{I} = 1$ for all $i, j$.[3] Intuitively, with exposure-based item utility, the item fairness constraint will cause the users' recommendation policies move from being concentrated on the most popular items, to a more uniform distribution over items. If the users in the population have diverse preferences, each item will already have an approximately uniform probability of being recommended, so that imposing item fairness constraints has a low cost.

In Figures 4 and 5 we examine the robustness of our empirical results in 3a and 3b respectively as we interpolate the item utility model between symmetry and exposure. In Figure 5 we see that user

---

[2]Of course, this need not be the platform's utility: in general the platform might receive utility $r_{ij}$ if user $i$ selects recommended item $j$.

[3]Again we could take $w_{ij}^{I} = b_j$ for some $b_j > 0$, but this would have equivalent normalized item utilities.

preference diversity indeed still improves the user-item fairness tradeoff when we change the item utility model. We also again see that the price of mis-estimation does not appear to increase with item fairness.

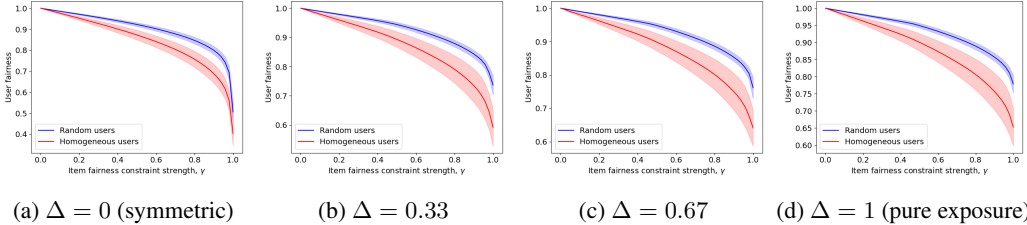

(a) $\Delta = 0$ (symmetric)    (b) $\Delta = 0.33$    (c) $\Delta = 0.67$    (d) $\Delta = 1$ (pure exposure)

Figure 4: Robustness of empirical findings without symmetry assumption: user-item fairness trade-offs and diversity as item utilities become less correlated with user utilities. Here, we take the user utilities $w^U$ to be derived from the arXiv recommendation engine similarity scores as in Figure 3a. For each plot the item utilities are a linear interpolation between the users' utilities and exposure. Formally, $w_{ij}^I = \Delta \cdot 1 + (1 - \Delta) \cdot w_{ij}^U$. When $\Delta = 0$, item and user utilities agree; when $\Delta = 1$ items derive utility solely from exposure.

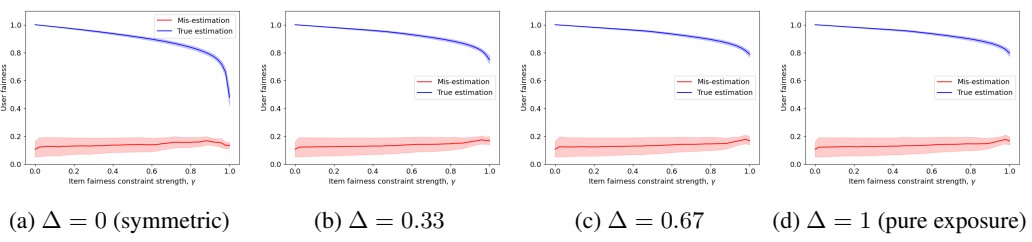

(a) $\Delta = 0$ (symmetric)    (b) $\Delta = 0.33$    (c) $\Delta = 0.67$    (d) $\Delta = 1$ (pure exposure)

Figure 5: Robustness of empirical findings without symmetry assumption: item fairness constraints still do not increase the price of mis-estimation empirically. Here, we take the user utilities $w^U$ to be derived from the arXiv recommendation engine similarity scores as in Figure 3b. For each plot the item utilities are a linear interpolation between the users' utilities and exposure. Formally, $w_{ij}^I = \Delta \cdot 1 + (1 - \Delta) \cdot w_{ij}^U$. When $\Delta = 0$, item and user utilities agree; when $\Delta = 1$ items derive utility solely from exposure.

## B    arXiv recommender empirical details

We prototype a recommender for preprints on arXiv, to illustrate our conceptual findings. We consider the cold start setting for items (papers), when they are newly uploaded to arXiv and so only have metadata but no associated interaction or citation data. For users (readers), we use as data the papers that *they* have published on arXiv in the past; we assume authors with identical names are the same author. We implement various natural language processing-based methods on the abstract text (for both items and the user's historical papers) to generate similarity scores (utility matrices) for users and items. We use the similarity scores to generate recommendations for each user, at various levels of user and item fairness constraints. We validate our approach by analyzing how citations correlate with the similarity score.

### B.1    Dataset and computation details

The original dataset was sourced from the public ArXiv Dataset available on Kaggle,[4] containing 1,796,911 articles. This dataset covers a wide range of scientific categories. Each entry in the dataset is characterized by features which include:

- **ID:** A unique identifier for each entry.
- **Authors:** Names and affiliations of the authors.

---

[4]Used under license CC0 Public Domain

- **Title:** The title of the paper.
- **Categories:** The scientific categories for the paper.
- **Abstract:** A brief summary of the paper.
- **Update Date:** The date of the latest update.

We further join this data with citation data from the Semantic Scholar API [25].[5] For each paper, we have both the papers that it cites and the papers that cite it.

We focus exclusively on entries classified under the 'Computer Science (CS)' category. We extract entries where the primary category designation was 'CS', and remove null and duplicate paper ID values. This filtering results in 177,323 entries.

This entire empirical workflow was run on a machine with 64 CPUs, 1 TB RAM, and 14 TB (non-SSD) disk. The estimated time was about 20 hours per week for 3 months, and the longest individual run was approximately 12 hours.

**Train and test data**   As training (to construct embeddings for users), we consider papers that those users published up to the end of 2019. Papers in 2020 are in the test set (the set available to be recommended). Papers after 2020 are used to evaluate recommendations (did the user cite a paper published in 2020).

The training dataset has 139,308 papers, and the test dataset has 14,307 papers, with the remaining being post-2020 papers. The training dataset has 178,260 distinct users with an average of approximately 2 papers per user. Figures 6a and 6b show the distribution of research paper subcategories between the train and test datasets. Figure 7 below shows the distribution of the number of papers per user in the training set on a logarithmic scale for the y-axis. The x-axis represents the number of papers per user, ranging from 0 to over 250.

Splitting the training and test data temporally both reflects practice and allows us to evaluate the recommendation model's effectiveness in predicting future citations. As detailed below, the model's success was measured by whether papers our model would have recommended to users in 2020 were in fact cited in their subsequent works. For further details, see below Section B.3 on the model evaluation.

## B.2   Recommendation models

Each recommendation approach has two design dimensions: (a) how we generate embeddings for each paper (papers to be recommended, and papers uploaded by users that will be used to construct user embeddings), and (b) how similarity scores are constructed once we have an embedding for each paper. Below, we evaluate each approach by their effectiveness in recommending papers that users are likely to cite in their future works.

**(a) Generating embeddings for each paper**   We use text-based analyses on the abstracts of each paper to generate paper embeddings. The first approach is TF-IDF, while the second employs Sentence Transformers.

**TF-IDF.** The first preprocessing step involved removing stopwords—common words with little informational value, such as "and" and "the"—to reduce noise and emphasize meaningful content. Next, a TF-IDF (Term Frequency-Inverse Document Frequency) vectorizer is applied to count term frequencies and scale them according to their rarity across the dataset, highlighting unique and informative words. The resulting frequency vector for each abstract is used as its embedding.

**Sentence Transformers.** The author-based recommendation model using Sentence Transformers leverages contextual embeddings from sentence-level representations. The preprocessing involves tokenizing the text (title, abstract, and categories) and generating embeddings using a pre-trained Sentence Transformer model, specifically the AllenAI SPECTER model [13],[6] available on Hugging Face.

---

[5]Used under the Semantic Scholar API License Agreement
[6]Licensed under the Apache License, Version 2.0

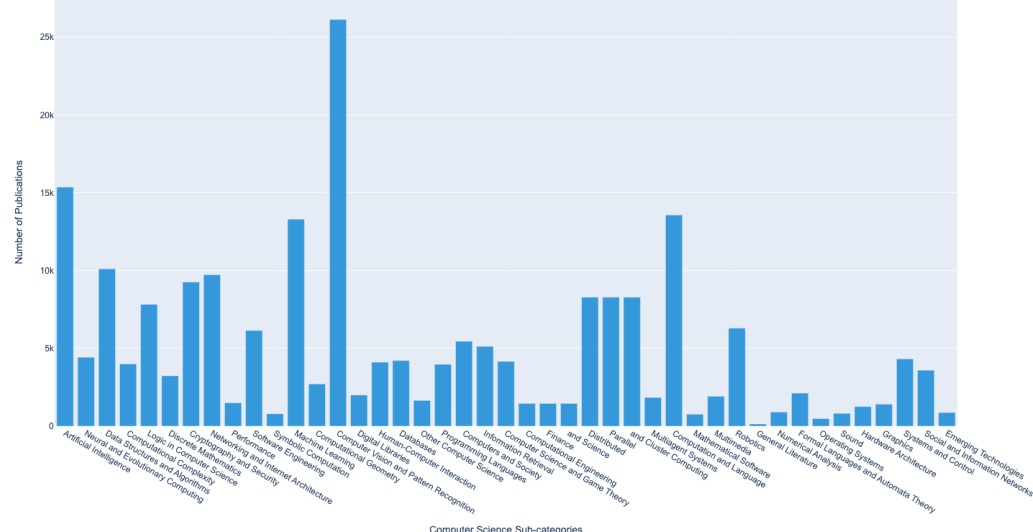

(a) Distribution of research paper publications in the train dataset over time

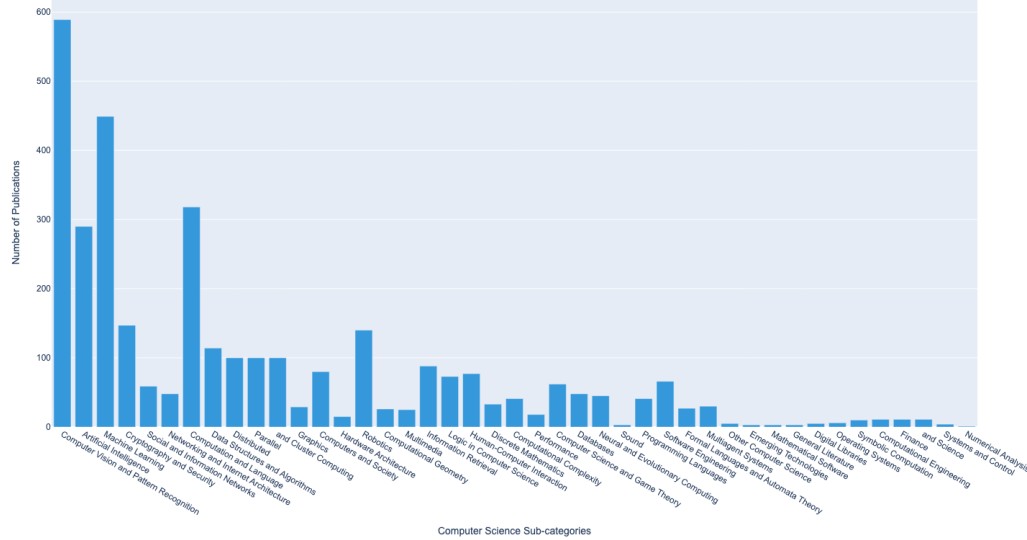

(b) Distribution of research paper publications in the test dataset over time

Figure 6: Distribution of research paper publications over time

**(b) Constructing similarity scores for each user-paper pair.** After constructing embeddings, we have an embedding for each paper in the potential recommendation set, and for each paper authored by a user in the training set. We construct user-paper similarity scores as follows. First, we compute the cosine similarity between each recommendation set embedding and each user's papers' embeddings. Then, the similarity score between each user and the recommendations set paper is constructed using one of the following approaches:

**Mean score.** We take the mean of the cosine similarities between the recommendation set paper and each of the papers by the user in the training set. This approach is equivalent to constructing a user embedding as the mean embedding of their uploaded papers.

**Max score.** The mean similarity score has been recognized as not capturing *diverse* interests that a user may have [20, 41] – for example, for a user who has published papers in two

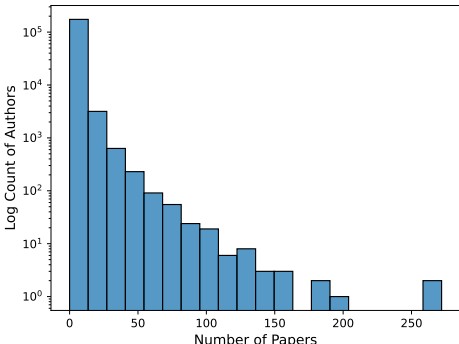

Figure 7: Distribution of the number of papers per user in the training set on a logarithmic scale

> different subject areas, a paper should be recommended to them if it matches *either* of their interests, as opposed to the average of those interests. Thus, we also construct user-paper similarity scores via the *max* similarity between any of the user's training set papers, and the recommended set paper.

Note that we also experimented with using dot products instead of cosine similarity; results are similar and omitted.

## B.3    Model evaluation method

We evaluate each of the four approaches (TF-IDF and sentence transformers, each with the max or the mean scores) using citation data for 1,128 users (authors) and 14,307 papers. For each user-paper pair, we use the following as outcome data:

**User cites paper in the future:**  Is the paper cited by the user in the future?

**Paper cites user:**  Does the paper cite the user already? We note that this data is technically available at the time of a hypothetical recommendation for a new paper, and so in theory could be used by a recommender. Thus, although metric does not directly measure the future behavior of the user, it helps in understanding the contextual alignment (determined just by natural language processing of the abstracts) of the recommended papers with the user's previous research or interests. Importantly, these references are not part of the data used to generate the similarity score.

In both cases, we consider the presence of citations as signifying a good recommendation.

For each recommendation approach, we calculate the relationship between this citation data and the following similarity score measures.

- **Similarity score:** The raw cosine similarity score.
- **Score percentile:** The percentile rank of the similarity score, where percentile is calculated *for each user*.
- **Normalized score:** The similarity score is normalized by subtracting the mean and dividing by the standard deviation of scores *for each user*.

A successful recommendation approach would have a strong relationship between text-based similarity scores and future citation outcomes.

## B.4    Evaluation results

We evaluate each of the 4 approaches. As summarized below, we find that the best performing approach is using the TF-IDF vectorizer to construct abstract embeddings, and then using the *max* similarity scores between any of their papers in the training set and each paper in the recommended

set. We find that this approach is effective at predicting future citations by the user, especially for a cold start recommender that uses only abstract textual information, and for such a sparse outcome as citations.

| Citation Type | No/Yes | Similarity Score | Score Percentile | Normalized Score |
|---|---|---|---|---|
| User cites paper | No | 0.041454 | 0.499737 | -0.001701 |
| | Yes | 0.123650 | 0.765008 | 1.514331 |
| Paper cites user | No | 0.041147 | 0.498860 | -0.006536 |
| | Yes | 0.138199 | 0.784410 | 1.582545 |

Table 2: Average similarity measures in the **Max score, TF-IDF** model, conditioned on citation presence for different types of citations.

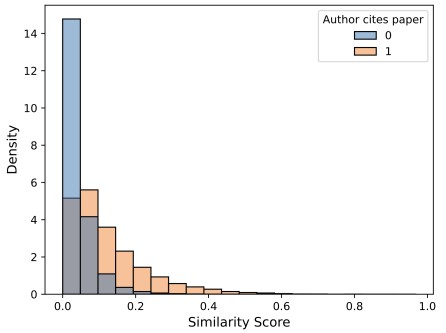

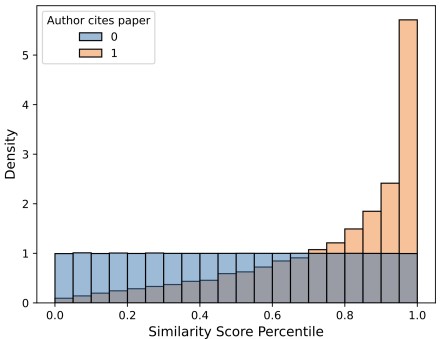

(a) Density plot of similarity scores grouped by citation presence.

(b) Density plot of similarity score percentiles grouped by citation presence.

Figure 8: $\Pr(\text{score}|\text{User cites paper})$, the distribution of the score for a user-paper pair, conditional on whether the user cites the paper in the future, for the **Max score, TF-IDF** model.

| Variable | Coefficient | Std. Err | z-value | P-value | Adjusted $R^2$ |
|---|---|---|---|---|---|
| Similarity score | 12.4100 | 0.058 | 212.178 | 0.000 | 0.08915 |
| Score percentile | 3.9218 | 0.035 | 111.194 | 0.000 | 0.05973 |
| Normalized Score | 0.3905 | 0.002 | 158.421 | 0.000 | 0.05159 |

Table 3: Logistic regression results for predicting whether user $i$ cites paper $j$ from the similarity score $w_{ij}$, for the **Max score, TF-IDF** model

**Max score, TF-IDF**   Table 2 shows the similarity measures described above averaged over all users and test papers, conditioned on whether or not a citation occurred between the user and paper (whether the user cites the paper, or vice versa). Figure 8 illustrates the distribution of similarity scores for papers that were cited (orange) versus those that were not cited (blue) by the user in the future. The density for cited papers is higher at higher levels of similarity scores compared to non-cited papers. Table 3 performing logistic regression between the user-paper score and whether the user cites the paper with a bias term ('User cites paper $\sim 1+$ score'), for each score measure. All measures suggest that our text-based recommendation scores are effective for predicting whether the user will cite the given paper. For example, Figure 8b shows that $\Pr(\text{Highest score bin}|\text{Author cites paper})$ is more than five times $\Pr(\text{Highest score bin}|\text{Author does not cite paper})$.

**Mean score, TF-IDF**   Table 4 shows the similarity measures described above averaged over all users and test papers, conditioned on whether or not a citation occurred between the user and paper (whether the author cites the paper, or vice versa). Figure 9 illustrates the distribution of similarity scores for papers that were cited (orange) versus those that were not cited (blue) by the user in the future. The density for cited papers is higher at higher levels of similarity scores compared to non-cited papers. Table 5 performing logistic regression between the user-paper score and whether the user cites the paper with a bias term ('User cites paper $\sim 1 +$ score'), for each score measure.

**Max score, Sentence transformer**

| Citation Type | No/Yes | Similarity Score | Score Percentile | Normalized Score |
|---|---|---|---|---|
| Author cites paper | No | 0.019405 | 0.499717 | -0.001799 |
| | Yes | 0.044938 | 0.783110 | 1.600929 |
| Paper cites author | No | 0.019323 | 0.498790 | -0.006737 |
| | Yes | 0.046283 | 0.801534 | 1.631049 |

Table 4: Average similarity measures in the **Mean score, TF-IDF** model, conditioned on citation presence for different types of citations.

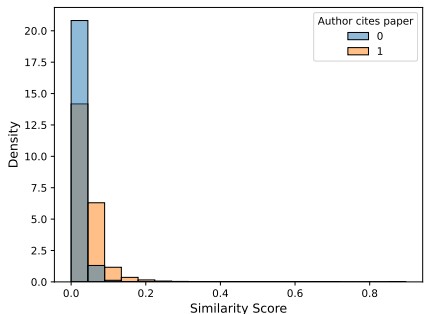

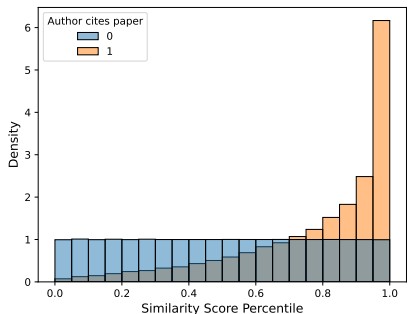

(a) Density plot of similarity scores grouped by citation presence.

(b) Density plot of similarity score percentiles grouped by citation presence.

Figure 9: $\Pr(\text{score}|\text{Author cites paper})$, the distribution of the score for a user-paper pair, conditional on whether the user cites the paper in the future, for the **Mean score, TF-IDF** model.

| Variable | Coefficient | Std. Err | z-value | P-value | Adjusted $R^2$ |
|---|---|---|---|---|---|
| Similarity score | 20.2122 | 0.131 | 154.835 | 0.000 | 0.04616 |
| Score percentile | 4.3535 | 0.037 | 116.516 | 0.000 | 0.06934 |
| Normalized Score | 0.4184 | 0.003 | 164.894 | 0.000 | 0.05815 |

Table 5: Logistic regression results for predicting whether user $i$ cites paper $j$ from the similarity score $w_{ij}$,, for the **Mean score, TF-IDF** model

| Citation Type | No/Yes | Similarity Score | Score Percentile | Normalized Score |
|---|---|---|---|---|
| Author cites paper | No | 0.666801 | 0.499821 | -0.00183 |
| | Yes | 0.784402 | 0.807105 | 1.30730 |
| Paper cites author | No | 0.666379 | 0.498851 | -0.005735 |
| | Yes | 0.794945 | 0.805592 | 1.251814 |

Table 6: Average similarity measures in the **Max score, Sentence transformer** model, conditioned on citation presence for different types of citations.

| Variable | Coefficient | Std. Err | z-value | P-value | Adjusted $R^2$ |
|---|---|---|---|---|---|
| Similarity score | 18.4557 | 0.250 | 73.695 | 0.000 | 0.1347 |
| Score percentile | 5.0122 | 0.098 | 51.161 | 0.000 | 0.08604 |
| Normalized Score | 1.2332 | 0.017 | 71.426 | 0.000 | 0.1076 |

Table 7: Logistic regression results for predicting whether user $i$ cites paper $j$ from the similarity score $w_{ij}$,, for the **Max score, Sentence transformer** model

Table 6 shows the similarity measures described above averaged over all users and test papers, conditioned on whether or not a citation occurred between the user and paper (whether the user cites the paper, or vice versa). Figure 10 illustrates the distribution of similarity scores for papers that were cited (orange) versus those that were not cited (blue) by the user in the future. The density for cited papers is higher at higher levels of similarity scores compared to non-cited papers. Table 7 shows the results of performing logistic regression between the user-paper score and whether the user cites the paper with a bias term ('User cites paper $\sim 1 + \text{score}$'), for each score measure.

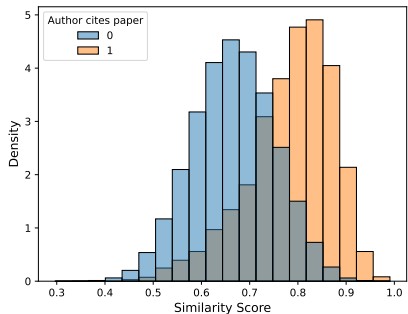
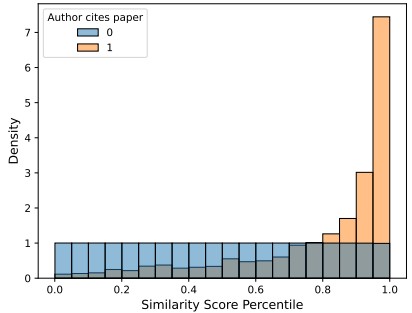

(a) Density plot of similarity scores grouped by citation presence.

(b) Density plot of similarity score percentiles grouped by citation presence.

Figure 10: $\Pr(\text{score}|\text{Author cites paper})$, the distribution of the score for a user-paper pair, conditional on whether the user cites the paper in the future, for the **Max score, Sentence transformer** model.

**Mean score, Sentence transformer**

| Citation Type | No/Yes | Similarity Score | Score Percentile | Normalized Score |
|---|---|---|---|---|
| User cites paper | No | 0.612678 | 0.499825 | -0.001767 |
| | Yes | 0.696291 | 0.803641 | 1.262704 |
| Paper cites user | No | 0.612397 | 0.498891 | -0.005418 |
| | Yes | 0.699609 | 0.796941 | 1.182465 |

Table 8: Average similarity measures in the **Mean score, Sentence transformer** model, conditioned on citation presence for different types of citations.

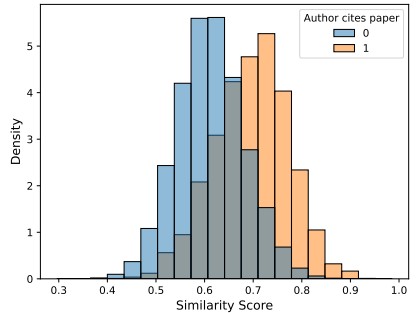

(a) Density plot of similarity scores grouped by citation presence.

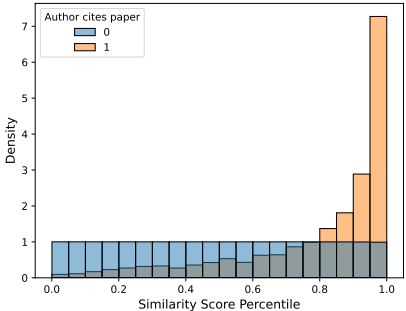

(b) Density plot of similarity score percentiles grouped by citation presence.

Figure 11: $\Pr(\text{score}|\text{User cites paper})$, the distribution of the score for a user-paper pair, conditional on whether the user cites the paper in the future, for the **Mean score, Sentence transformer** model.

| Variable | Coefficient | Std. Err | z-value | P-value | Adjusted $R^2$ |
|---|---|---|---|---|---|
| Similarity score | 16.2482 | 0.246 | 66.148 | 0.000 | 0.09085 |
| Score percentile | 4.9088 | 0.097 | 50.820 | 0.000 | 0.08377 |
| Normalized Score | 1.2297 | 0.018 | 69.188 | 0.000 | 0.1026 |

Table 9: Logistic regression results for predicting whether user $i$ cites paper $j$ from the similarity score $w_{ij}$,, for the **Mean score, Sentence transformer** model

Table 8 shows the similarity measures described above averaged over all users and test papers, conditioned on whether or not a citation occurred between the user and paper (whether the user cites the paper, or vice versa). Figure 11 illustrates the distribution of similarity scores for papers that were cited (orange) versus those that were not cited (blue) by the user in the future. The density for cited papers is higher at higher levels of similarity scores compared to non-cited papers. Table 9 shows the results of performing logistic regression between the user-paper score and whether the user cites the paper with a bias term ('User cites paper $\sim 1 + $ score'), for each score measure.

Altogether, each model performs fairly well; the model using sentence transformer embeddings and the max similarity score among each author's papers appears to perform the best overall.

## C  Proof of Section 3 results

In this section, we provide complete proofs for Proposition 1 and Proposition 2. In this and all other proofs in the appendix, we ignore the dependence on the utility matrix $w$ in the user and item utility and fairness functions when clear.

The following result will be broadly useful throughout the rest of the proofs.

**Lemma 1.** *Since $w_{ij} > 0$ for all $j$, $I_{\min}^* > 0$.*

*Proof.* Let $\rho_{ij} = \frac{1}{n}$. Then $I_j(\rho) > 0$ for all $j$ so $I_{\min}(\rho) > 0$ and $I_{\min}^* = \max_\rho I_{\min}(\rho) > 0$. $\quad\square$

First, recall the statement of Proposition 1.

**Proposition 1.** *Suppose that for a set of recommendation policies $\mathcal{S} \subseteq \Delta_{n-1}^m$,*

*(i) $\mathcal{S}$ can be described by a finite set of linear constraints*

*(ii) There exists an optimal solution $\rho^*$ to Problem (2) such that $\rho^* \in \mathcal{S}$*

*(iii) $\rho^*$ is the unique feasible solution to Problem (2) in $\mathcal{S}$*

*Then, finding an optimal solution $\rho^*$ to Problem (2) can be reduced to solving a linear program $\mathcal{L}$.*

*Proof.* We first need the following result, which states that the feasible set of 1 can be expressed as a linear program. We will prove this result below.

**Lemma 2.** *Let $\mathcal{F}$ denote the feasible set of Problem 1,*

$$\mathcal{F} \triangleq \arg \max_{\rho \in \Delta_{n-1}^m} I_{\min}(\rho) \triangleq \arg \max_{\rho \in \Delta_{n-1}^m} \min_j I_j(\rho). \tag{3}$$

*Then $\mathcal{F}$ is the solution set of the linear program*

$$\mathcal{F} = \arg \max_{\rho \in \Delta_{n-1}^m, \lambda} \quad \lambda$$
$$\text{subject to} \quad I_j(\rho) = \lambda \quad \forall j. \tag{4}$$

Now, condition (iii) on $\mathcal{S}$ implies that there is a unique $\rho^*$ in $\mathcal{S} \cap \mathcal{F}$. Thus

$$\rho^* = \mathcal{F} \cap \mathcal{S} = \arg \max_{\rho \in \Delta_{n-1}^m \cap \mathcal{S}, \lambda} \quad \lambda$$
$$\text{subject to} \quad I_j(\rho) = \lambda \quad \forall j, \tag{5}$$

We can add $\mathcal{S}$ as a constraint in Problem 4 because we know that there is at least one solution to Problem 4 in $\mathcal{S}$ – namely, $\rho^*$ – so this additional constraint will not change the optimal objective value, but will constrain the set of optimal solutions to be in $\mathcal{S}$, as desired.

By condition (ii), $\rho^*$ is an optimal solution for Problem 1; by condition (i), $\mathcal{S}$ can be described by a finite set of linear constraints, and thus Problem 5 is a linear program; call this $\mathcal{L}$. $\square$

We now show Lemma 2.

**Lemma 2.** *Let $\mathcal{F}$ denote the feasible set of Problem 1,*

$$\mathcal{F} \triangleq \arg \max_{\rho \in \Delta_{n-1}^m} I_{\min}(\rho) \triangleq \arg \max_{\rho \in \Delta_{n-1}^m} \min_j I_j(\rho). \tag{3}$$

*Then $\mathcal{F}$ is the solution set of the linear program*

$$\mathcal{F} = \arg \max_{\rho \in \Delta_{n-1}^m, \lambda} \quad \lambda$$
$$\text{subject to} \quad I_j(\rho) = \lambda \quad \forall j. \tag{4}$$

*Proof.* To show this, we prove by contradiction that every optimal policy gives each item $j$ the same normalized utility; that is, we show that if the policy $\rho$ is an optimal solution of Problem 3, then there is some $\lambda$ such that for all $j$, $I_j(\rho) = \lambda$, and in this case $\lambda = \min_j I_j(\rho)$. This implies that any solution $\rho$ to Problem 3 is a feasible solution of 4. Since these two problems have the same objective, and the additional constraint in Problem 4 does not eliminate any solutions, the two problems have the same solution set.

To show that all items have the same normalized utility at the optimum, we suppose the contrary and produce a contradiction. Suppose that the policy $\rho$ maximizes $\min_j I_j(\rho)$, but there is some $j$ such that $I_j(\rho) > I_{\min}^*$. We will show that this means we can construct $\rho'$ such that $I_{\min}(\rho') > I_{\min}^*$.

First, note that it is impossible that $\rho_{ij} = 0$ for all $i$, otherwise $I_{\min}^* = I_j(\rho) = 0$. However, by Lemma 1, $I_{\min}^* > 0$. Thus for some $i$, $\rho_{ij} > 0$.

Define $\mathcal{J} = \{j' : I_{j'}(\rho) = I_{\min}^*\}$, and let $j' \in \mathcal{J}$. Since $\rho_{ij} > 0$ and $\sum_k \rho_{ik} = 1$, $\rho_{ij'} < 1$. Pick $\epsilon > 0$ such that $\rho'_{ij} := \rho_{ij} - \epsilon > 0$, $\rho'_{ij'} := \rho_{ij'} + \epsilon < 1$, and $I_j(\rho') > I_{\min}^*$. Since $I_j$ is linear in the recommendation policy, this is always possible.

Now, repeat this process for all $j' \in \mathcal{J}$, to obtain a final recommendation policy $\rho'$ in which $I_\ell(\rho') > I_{\min}^*$ for all $\ell$ and thus $I_{\min}(\rho') > I_{\min}^*$. This contradicts the optimality of $I_{\min}^*$. $\square$

Now, we prove Proposition 2.

**Proposition 2.** $\mathcal{S}_{symm}$ *satisfies conditions (i) and (ii) in Proposition 1. Furthermore, solutions* $\rho \in \mathcal{S}_{symm}$ *to Problem* (2) *have a sparse structure:*

- *If $\rho$ is a basic feasible solution to the linear program $\mathcal{L}$ in Proposition 1, then there are at most $n + K - 1$ type-item pairs $(k, j)$ such that $\rho_{kj} > 0$ (out of $nK$ possible pairs).*

- *If $\rho$ is also optimal, then there are at most $K - 1$ items that are ever recommended to more than one type of user, i.e., where $\rho_{kj}, \rho_{k'j} > 0$ for $k \neq k'$.*

*Proof.* We will first prove that $\mathcal{S}_{symm}$ satisfies conditions (i) and (ii), and then show sparsity.

**Part 1.** Recall that for fixed $w$,

$$\mathcal{S}_{symm} = \{\rho : \rho_i = \rho_{i'} \text{ if } w_i = w_{i'}\} = \bigcap_{i, i' : w_i = w_{i'}} \{\rho : \rho_i = \rho_{i'}\}$$

which is a finite set of linear constraints, so condition (i) holds.

Now, we show condition (ii), that always there is some $\phi^* \in \mathcal{S}_{symm}$ that is an optimal solution to Problem 1. Let $\rho$ be an arbitrary optimal solution to Problem 1. Then $\rho$ is also feasible: $I_{\min}(\rho) = I_{\min}^*$.

We first define a piece of convenient notation: if two users $i, i'$ share the same utility vector $w_i = w_{i'}$, we say that they have the same type $\tau(i) = \tau(i')$. If $\tau(i) = \tau(i') := \tau$, we write $w_{ij} = w_{i'j} := w_{\tau j}$

Let

$$\phi_{ij} = \frac{1}{|\{i' : \tau(i') = \tau(i)\}|} \sum_{i' : \tau(i') = \tau(i)} \rho_{i'j}$$

for all $i, j$. We must show that:

- $\phi_{ij} = \phi_{i'j}$ for all $i, i'$ where $\tau(i) = \tau(i')$,

- $\phi$ is a valid set of recommendation probabilities,

- $\phi$ is feasible: $I_{\min}(\phi) = I_{\min}^*$, and

- $\phi$ is optimal: $U_{\min}(\phi) \geq U_{\min}(\rho) = U_{\min}^*$.

Clearly $\phi_{ij} = \phi_{i'j}$ for all $i, i'$ where $\tau(i) = \tau(i')$, since $\phi_{ij}$ depends only on $i$ through $\tau(i)$.

For each $i$,

$$\sum_j \phi_{ij} = \frac{1}{|\{i' : \tau(i') = \tau(i)\}|} \sum_{i' : \tau(i') = \tau(i)} \sum_j \rho_{i'j} = \frac{1}{|\{i' : \tau(i') = \tau(i)\}|} \sum_{i' : \tau(i') = \tau(i)} 1 = 1.$$

Moreover, for all $i, j$, $0 \leq \min_{r,s} \rho_{rs} \leq \phi_{ij} \leq \max_{r,s} \rho_{rs} \leq 1$. Thus $\phi \in \Delta_{n-1}^m$.

To show that $I_{\min}(\phi) = I_{\min}^*$, notice that intuitively, when we move from $\rho$ to $\phi$, we redistribute the probability mass assigned to an item $j$ among users of the same type. Since all of these users generate the same value for item $j$, there should be no change in item $j$'s expected utility. Formally,

$$\begin{aligned}
I_{\min}(\phi) &= \min_j \frac{\sum_\tau w_{\tau j} \sum_{i:\tau(i)=\tau} \phi_{ij}}{\sum_\tau w_{\tau j} \sum_{i:\tau(i)=\tau} 1} \\
&= \min_j \frac{\sum_\tau w_{\tau j} \sum_{i:\tau(i)=\tau} \rho_{ij}}{\sum_\tau w_{\tau j} \sum_{i:\tau(i)=\tau} 1} \\
&= I_{\min}(\rho) \\
&= I_{\min}^*. && (\rho \text{ is optimal})
\end{aligned}$$

Finally, to show that $U_{\min}(\phi) \geq U_{\min}(\rho)$, we observe that if users with the same values are given different recommendation probabilities, some user will be worst off and have expected value lower than the average expected value over all users in that type. By averaging, we bring all users' expected

values to the average expected value across the type, increasing the expected value for the previously worst off user. Formally, for each type $\tau$, let $i(\tau) \in \arg\min_{i:\tau(i)=\tau} U_i(\rho)$. Then

$$U_{i(\tau)}(\rho) \leq U_{i'}(\rho) \text{ for all } i' : \tau(i') = \tau$$

$$\implies \frac{\sum_{j=1}^n w_{\tau j}\rho_{i(\tau)j}}{\max_j v_j} \leq \frac{\sum_{j=1}^n w_{\tau j}\rho_{i'j}}{\max_j v_j} \text{ for all } i' : \tau(i') = \tau$$

$$\implies |\{i' : \tau(i') = \tau\}|\frac{\sum_{j=1}^n w_{\tau j}\rho_{i(\tau)j}}{\max_j v_j} \leq \sum_{i':\tau(i')=\tau} \frac{\sum_{j=1}^n w_{\tau j}\rho_{i'j}}{\max_j v_j}$$

$$\implies \frac{\sum_{j=1}^n w_{\tau j}\rho_{i(\tau)j}}{\max_j v_j} \leq \frac{\sum_{j=1}^n w_{\tau j}\phi_{i'j}}{\max_j v_j} \text{ for all } i' : \tau(i') = \tau \qquad \text{(definition of } \phi)$$

$$\implies \min_{i:\tau(i)=\tau} U_i(\rho) \leq \min_{i:\tau(i)=\tau} U_i(\phi)$$

Then

$$U_{\min}(\rho) = \min_\tau \min_{i:\tau(i)=\tau} U_i(\rho) \leq \min_\tau \min_{i:\tau(i)=\tau} U_i(\phi) = U_{\min}(\phi)$$

and we have shown that $U_{\min}(\phi) \geq U_{\min}(\rho)$.

**Part 2.** Now we show the sparsity of solutions to the linear program $\mathcal{L}$ in Proposition 1, that is, Problem 5 defined above in the proof of Proposition 1. Given that $\rho \in \mathcal{S}_{\text{symm}}$, we can rewrite the linear program as

$$\phi = \arg\max_{\rho \in \Delta_{n-1}^K, \lambda} \quad \lambda$$
$$\text{subject to} \quad I_j(\rho) = \lambda \quad \forall j,$$

which is the same as

$$\phi = \arg\max_{\rho, \lambda} \quad \lambda$$
$$\text{subject to} \quad I_j(\rho) = \lambda \quad \forall j,$$
$$\sum_j \rho_{kj} = 1 \quad \forall k,$$
$$\rho_{kj} \geq 0 \quad \forall k, j,$$

where $I_j(\rho)$ is shorthand for $I_j(\rho')$ where $\rho' \in \Delta_{n-1}^m$ is $\rho \in \Delta_{n-1}^K$ expressed in terms of users rather than types. This is a linear program with $nK + 1$ variables, and $n + K$ equality constraints. Then for any basic feasible solution $\rho$ of Problem 5, there must be $nK + 1$ linearly independent active constraints (see Definition 2.9 in Bertsimas and Tsitsiklis [7]). This means that of the $nK$ constraints $\{\rho_{kj} \geq 0\}_{k,j}$, at least $nK + 1 - (n + K)$ must be binding. Equivalently, there can be at most $nK - (nK + 1 - (n + K)) = n + K - 1$ values of $k, j$ such that $\rho_{kj} > 0$.

If $\rho$ is also an optimal solution, then by Lemma 1 there must be some $k$ such that $\rho_{kj} > 0$ for each $j$, which takes up $n$ non-zero values. Only $K - 1$ non-zero values remain to be assigned, so at most $K - 1$ items $j$ have $\rho_{kj}, \rho_{k'j} > 0$ for two types $k, k'$. $\qquad \square$

# D   Proof of Theorem 3

Recall Theorem 3 and its setting. We let $v_1 > v_2 > ... > v_n$ and suppose that there are two user types. A user $i$ either has value $w_{ij} = v_j$ for all $j$ or $w_{ij} = v_{n-j+1}$ for all $j$, corresponding to types 1 and 2 respectively. That is, the two types have opposite preferences. We let $\alpha$ be the proportion of type 1 users in the population, out of a fixed population of $m$ users.

**Theorem 3.** $\pi_{U|I}^F(\alpha)$ *is decreasing in $\alpha$ for $0 < \alpha \leq 1/2$, and increasing in $\alpha$ for $1/2 \leq \alpha < 1$.*

## D.1   Main proof

*Proof.* The proof will proceed in two parts. First, we will reduce the problem to finding a solution for a linear program using the framework from Proposition 1. Then, we will find a solution to the linear program. In this proof, we will use a series of intermediate results, which we prove in the following section. The first such result is Lemma 3.

**Lemma 3.** *For all $0 < \alpha < 1$, $U^*_{\min}(\alpha) = 1$.*

By Lemma 3, $U^*_{\min}(\alpha) := U^*_{\min}$ is constant in $\alpha$, so $\pi^F_{U|I}(\alpha) = {(U^*_{\min} - U^*_{\min}(1,\alpha))}/{U^*_{\min}}$ and it is enough to show $U^*_{\min}(1,\alpha)$ increases in $\alpha$.

Recall the definition of $\mathcal{S}_{\text{symm}}$:

$$\{\rho : \rho_i = \rho_{i'} \text{ if } w_i = w_{i'}\}.$$

By Proposition 2, we know that $\mathcal{S}_{\text{symm}}$ is a satisfies conditions (i) and (ii) on $\mathcal{S}$ for Proposition 1. Thus by Proposition 1, it is sufficient to find $\phi$ such that

$$\phi \in \arg\max_{\rho \in \mathcal{S} \cap \Delta^m_{n-1}, \lambda} \lambda$$
$$\text{subject to} \quad I_j(\rho) = \lambda \quad \forall j,$$

and show that this $\phi$ is unique, where the linear program is derived from the proof of Proposition 1.

Since each user in our population only has one of two possible value vectors, $w_i = (v_1, ..., v_n)$ or $w_i = (v_n, ...., v_1)$, we can identify $\mathcal{S}_{\text{symm}}$ with $\Delta^2_{n-1}$, and identify $\rho \in \mathcal{S}_{\text{symm}}$ with a pair $x, y$. Expanding the expression for $I_j(\rho)$ and making explicit the dependence on $\alpha$, we see that

$$\begin{aligned}
I_j(\rho, \alpha) &= \frac{\sum_i \rho_{ij} w_{ij}(\alpha)}{\sum_i w_{ij}(\alpha)} \\
&= \frac{\sum_{i:\tau(i)=1} \rho_{ij} v_j + \sum_{i:\tau(i)=2} \rho_{ij} v_{n-j+1}}{\sum_{i:\tau(i)=1} v_j + \sum_{i:\tau(i)=2} v_{n-j+1}} \\
&= \frac{m\alpha x_j v_j + m(1-\alpha) y_j v_{n-j+1}}{m\alpha v_j + m(1-\alpha) v_{n-j+1}} \\
&= q_j(\alpha) x_j + (1 - q_j(\alpha)) y_j
\end{aligned}$$

where

$$q_j(\alpha) := \frac{\alpha v_j}{\alpha v_j + (1-\alpha) v_{n-j+1}}.$$

Note that for all $j$ and $0 < \alpha < 1$, we have $0 < q_j, 1 - q_j < 1$. We can thus simplify

$$\phi = \arg\max_{x,y \in \Delta^2_{n-1}, \lambda} \lambda \tag{6}$$
$$\text{subject to} \quad q_j(\alpha) x_j + (1 - q_j(\alpha)) y_j = \lambda \quad \forall j,$$

First, we will show uniqueness and find a closed form to Problem 6 in Lemmas 4 and 5. Then, we will use this closed forms to show that $U^*_{\min}(1,\alpha)$ is increasing for $\alpha < 1/2$. By symmetry, this implies that $U^*_{\min}(1,\alpha)$ must be decreasing for $\alpha > 1/2$ and we will be done.

**Lemma 4.** *Problem 6 has a unique solution $(x, y, \lambda)$. Moreover, the solution is sparse: there is some $1 \le t \le n$ such that for $j > t$, $x_j = 0$, and for $j < t$, $y_j = 0$.*

Thus, to solve our original problem (Problem 1) we merely need to evaluate the user fairness $U_{\min}$ of $x, y$ solving Problem 6 and show that this is increasing. We will do this in the remainder of the proof.

For each $\alpha$, let $x, y$ be the solution to the corresponding Problem 6, and define[7]

$$t(\alpha) := \max\{j : x_j(\alpha) > 0\}.$$

**Lemma 5.** *Let $0 < \alpha < 1$, $t := t(\alpha)$. Define*

$$L_t = \sum_{j<t} \frac{1}{q_j}, \quad R_t = \sum_{j>t} \frac{1}{1 - q_j}.$$

---

[7]While there is a single optimal solution $x, y$ to Problem 6 for each $\alpha$, there may be two $t$ that satisfy Lemma 4. In particular, this occurs when for each $j$, $x_j = 0$ or $y_j = 0$. By contrast, $t(\alpha)$ is unique.

*Then*

$$I^*_{\min} = \frac{1}{1 + q_t L_t + (1 - q_t)R_t},$$

*and*

$$x_j = \begin{cases} \frac{I^*_{\min}}{q_j}, & j < t \\ 1 - \frac{L_t}{1 + q_t L_t + (1 - q_t)R_t}, & j = t \\ 0, & j > t \end{cases}, \quad y_j = \begin{cases} 0, & j < t \\ 1 - \frac{R_t}{1 + q_t L_t + (1 - q_t)R_t}, & j = t \\ \frac{I^*_{\min}}{1 - q_j}, & j > t \end{cases}.$$

Note that for $t \neq t(\alpha)$, $x, y$ above may not be valid probability vectors.

**Lemma 6.** *For $0 < \alpha < 1/2$, if $\rho^*$ is the optimal recommendation policy, then $U_i(\rho^*, \alpha) \geq U_{i'}(\rho^*, \alpha)$ for $\tau(i) = 1$, $\tau(i') = 2$.*

By Lemma 6, $U^*_{\min}(\alpha) = U_i(\alpha)$ for $i$ with $\tau(i) = 2$. Then

$$U^*_{\min}(1, \alpha) = U_i(\alpha)$$

$$= \frac{1}{\max_j v_j} \sum_{j=1}^{n} y_j v_{n-j+1}$$

$$= \frac{1}{\max_j v_j} \left( y_{t(\alpha)} v_{n-t(\alpha)+1} + \sum_{j>t(\alpha)} y_j v_{n-j+1} \right)$$

$$= \frac{1}{\max_j v_j} \left( \left( 1 - \sum_{j>t(\alpha)} \frac{I^*_{\min}(\alpha)}{1 - q_j(\alpha)} \right) v_{n-t(\alpha)+1} + \sum_{j>t(\alpha)} \frac{I^*_{\min}(\alpha)}{1 - q_j(\alpha)} v_{n-j+1} \right)$$

$$= \frac{1}{\max_j v_j} \left( v_{n-t(\alpha)+1} + \sum_{j>t(\alpha)} \frac{I^*_{\min}(\alpha)}{1 - q_j(\alpha)} (v_{n-j+1} - v_{n-t(\alpha)+1}) \right)$$

The following results will enable us to conclude that $U^*_{\min}(1, \alpha)$ is increasing for $\alpha < 1/2$.

**Lemma 7.** $t(\alpha)$ *is weakly increasing.*

**Lemma 8.** $I^*_{\min}(\alpha)$ *is increasing for $\alpha \leq 1/2$.*

**Lemma 9.** $q_j(\alpha)$ *strictly increases as $\alpha$ increases and strictly decreases as $j$ increases.*

Since $t(\alpha)$ is increasing and $v_j$ decreases in $j$, $v_{n-t(\alpha)+1}$ is increasing in $\alpha$. Also, $I^*_{\min}(\alpha)$ is increasing for $\alpha < 1/2$, and $q_j(\alpha)$ is increasing, so $\frac{I^*_{\min}(\alpha)}{1 - q_j(\alpha)}$ is increasing. For $j > t(\alpha)$, $v_{n-j+1} - v_{n-t(\alpha)+1} > 0$. Thus, $U^*_{\min}(1, \alpha)$ is increasing. $\qquad\square$

### D.2 Supporting lemma proofs

We will first prove Lemma 9, as it is a simple result that we will use extensively.

**Lemma 9.** $q_j(\alpha)$ *strictly increases as $\alpha$ increases and strictly decreases as $j$ increases.*

*Proof.* Recall the definition of $q_j(\alpha)$:

$$q_j(\alpha) = \frac{\alpha v_j}{\alpha v_j + (1 - \alpha)v_{n-j+1}}.$$

We can rewrite this as

$$q_j(\alpha) = \frac{1}{1 + \frac{(1-\alpha)v_{n-j+1}}{\alpha v_j}} = \frac{1}{1 + (\frac{1}{\alpha} - 1)\frac{v_{n-j+1}}{v_j}},$$

which is strictly increasing in $\alpha$ since $v_{n-j+1}/v_j > 0$.

Since $v_j$ is decreasing in $j$, $v_{n-j+1}/v_j$ is increasing in $j$. Then since $1/\alpha > 1$, $\left(\frac{1}{\alpha} - 1\right)\frac{v_{n-j+1}}{v_j}$ is increasing, so $q_j(\alpha)$ is decreasing. $\qquad\square$

**Lemma 3.** *For all $0 < \alpha < 1$, $U^*_{\min}(\alpha) = 1$.*

*Proof.* Simply put, the argument below says that since there are no total utility or item fairness constraints, we can simply maximize utility for each user individually. When we do this, each user gets a utility ratio of 1. Formally, recall that

$$U^*_{\min}(\alpha) = \max_{\rho \in \Delta^m_{n-1}} \min_i \frac{\sum_{j=1} \rho_{ij} w_{ij}(\alpha)}{\max_j w_{ij}(\alpha)}$$

We can pick $x_i$ for each user $i$ independently, so this problem becomes

$$U^*_{\min}(\alpha) = \min_i \frac{1}{\max_j w_{ij}} \max_{\rho_i \in \Delta_{n-1}} \sum_{j=1} x_{ij} w_{ij}(\alpha).$$

We know that $\sum_{j=1} \rho_{ij} w_{ij}$ is maximized by putting all probability mass of $\rho_i$ on the coordinate of $w_i$ with the highest value, yielding expected value $\max_j w_{ij}$. Then

$$U^*_{\min}(\alpha) = \min_i \frac{\max_j w_{ij}}{\max_j w_{ij}} = 1.$$

$\qquad\square$

**Lemma 4.** *Problem 6 has a unique solution $(x, y, \lambda)$. Moreover, the solution is sparse: there is some $1 \le t \le n$ such that for $j > t$, $x_j = 0$, and for $j < t$, $y_j = 0$.*

*Proof.* Let $\alpha$ be fixed; we will suppress the dependence on $\alpha$ for the remainded of the proof. Recall that Problem 6 is a linear program, and we can therefore rely on known facts about the structure of solutions to linear programs to prove this result.

**Part 1** First, we show that at least $n-1$ of $\{x_j, y_j\}_j$ must be zero in every basic feasible solution.

By Proposition 2, since $K = 2$, at most $n + 2 - 1 = n + 1$ of $\{x_j, y_j\}_j$ can be non-zero and thus at least $n-1$ of these are zero.

**Part 2** Now, we show that every optimal basic feasible solution has the form above.

If $x, y, \lambda$ define an optimal basic feasible solution and $x_j = 0$, then if $i > j$, $x_i = 0$. To see this, suppose this is not the case, that is, there is some $j < i$ such that $x_i := c > 0$ but $x_j = 0$. We will show that we can form $x', y'$ with $\min_j I_j(x', y') > \lambda$, which means that the policy $\rho'$ formed by giving recommendation policy $x'$ to all users of type 1 and recommendation policy $y'$ to all users of type 2, is a better solution to Problem 1 than the policy $\rho$ defined by $x, y$, which means $\rho$ is not optimal and we have a contradiction.

Since $x, y$ are feasible, we must have $I_i(x, y) = I_j(x', y') = \lambda$, so

$$q_i c + (1 - q_i)y_i = (1 - q_j)y_j. \tag{7}$$

Define $x', y'$ as follows:

$$x'_\ell = \begin{cases} c - \epsilon_1, & \ell = j \\ 0, & \ell = i \\ x_\ell + \frac{\epsilon_1}{n-2}, & \text{otherwise} \end{cases},$$

$$y'_\ell = \begin{cases} y_j - \left(\frac{q_i c}{1-q_i} + \epsilon_2\right), & \ell = j \\ y_i + \left(\frac{q_i c}{1-q_i} + \epsilon_2\right), & \ell = i \\ y_\ell, & \text{otherwise} \end{cases}.$$

Intuitively, to define $x'$ we move all probability mass away from $i$ to $j$. To define $y'$, we move sufficient mass from $j$ to $i$ to offset the decrease in value to item $i$ from changing $x$ to $x'$. In order to *strictly* increase $I_\ell$ for all items $\ell$, in $x'$ we also move $\epsilon_1$ mass to the other items.

These will be valid probability vectors for $\epsilon_1, \epsilon_2 > 0$ small enough; for $x'$ this is clear. For $y'$, we must show that $y_j > \frac{q_i c}{1 - q_i}$, and $y_i + \frac{q_i c}{1 - q_i} < 1$. Rearranging Equation 7 and noticing that $q_j > q_i$ by Lemma 9,

$$y_i + \frac{q_i c}{1 - q_i} = \frac{(1 - q_j) y_j}{1 - q_i} < y_j \leq 1.$$

Moreover,

$$
\begin{aligned}
y_j - \frac{q_i c}{1 - q_i} &= \frac{q_i c + (1 - q_i) y_i}{1 - q_j} - \frac{q_i c}{1 - q_i} && \text{(Equation 7)} \\
&> \frac{q_i c + (1 - q_i) y_i}{1 - q_i} - \frac{q_i c}{1 - q_i} && (q_j > q_i) \\
&= y_i. \\
&\geq 0
\end{aligned}
$$

Now, we can show that $I_{\min}(x', y') > I_{\min}(x, y) = I^*_{\min}$, which is a contradiction. To do this, it is sufficient to show that $I_\ell(x', y') > I_\ell(x, y)$ for each $\ell$.

- $\ell \notin \{i, j\}$: Since $x'_\ell > x_\ell$, $y'_\ell = y_\ell$, $I_\ell(x', y') > I_\ell(x, y)$.

- $\ell = i$:

$$
\begin{aligned}
I_i(x', y') &= (1 - q_i) \left( y_i + \frac{q_i c}{1 - q_i} + \epsilon_2 \right) \\
&= (1 - q_i) y_i + q_i \lambda + \epsilon_2 \\
&= I_i(x, y) + \epsilon_2
\end{aligned}
$$

- $\ell = j$:

$$
\begin{aligned}
I_j(x', y') &= q_j(\lambda - \epsilon_1) + (1 - q_j) \left( y_j - \frac{q_i \lambda}{1 - q_i} - \epsilon_2 \right) \\
&= (1 - q_j) y_j + \lambda \left( q_j - q_i \frac{1 - q_j}{1 - q_i} \right) - \epsilon_1 q_j - \epsilon_2 (1 - q_j) \\
&= I_j(x, y) + \lambda \left( q_j - q_i \frac{1 - q_j}{1 - q_i} \right) - \epsilon_1 q_j - \epsilon_2 (1 - q_j)
\end{aligned}
$$

Since $q_j > q_i$, $q_j - q_i \frac{1 - q_j}{1 - q_i} > 0$, and thus $I_j(x', y') > I_j(x, y)$ if $\epsilon_1, \epsilon_2$ are small enough.

Since we did not use the fact that $\alpha < 1/2$ here, a symmetric argument shows that we cannot have $y_j = 0$, $y_i > 0$, and $i < j$,

Let $k$ be the number of indices $j$ such that $x_j > 0$. The above arguments together imply that $x_j = 0$, $j > k$. There are then at least $n - 1 - (n - k) = k - 1$ indices $j$ such that $y_j = 0$; the argument above implies that these zeroes are on the lowest possible indices, that is, $y_j = 0$, $j < k$. Thus if we take $t = k$, we have that for $j > t$, $x_j = 0$, and for $j < t$, $y_j = 0$.

**Part 3** Finally, we show that there is only a single optimal solution, which is a basic feasible solution of this form.

Suppose that $x', y'$ is another optimal basic feasible solution; we will show that $x' = x, y' = y$. Denote $t(x, y) = \max\{j : x_j > 0\}$.

- If $t(x, y) = t(x', y')$, then:

– For $j < t$, $y_j = y'_j = 0$, and

$$
\begin{aligned}
q_j x_j &= I_j(x, y) && \text{(Lemma 5)} \\
&= I_{\min}(x, y) && \text{(feasibility)} \\
&= I_{\min}(x', y') && \text{(optimality)} \\
&= I_j(x', y') && \text{(feasbility)} \\
&= q_j x'_j, && \text{(Lemma 5)}
\end{aligned}
$$

so $x_j = x'_j$.

– Similarly for $j > t$, $x_j = x'_j = 0$, and

$$
(1 - q_j)y_j = I_j(x, y) = I_{\min}(x, y) = I_{\min}(x', y') = I_j(x', y') = (1 - q_j)y'_j,
$$

so $y'_j = y_j$.

– Finally, this means $y_t = 1 - \sum_{j>t} y_j = 1 - \sum_{j>t} y'_j = y'_t$, and $x_t = 1 - \sum_{j<t} x_j = 1 - \sum_{j<t} x'_j = x'_t$.

• Otherwise, without loss of generality let $t(x', y') > t(x, y) := t$. For $j < t$,

$$
q_j x_j = I_j(x, y) = I_j(x', y') = q_j x'_j,
$$

so $x'_j = x_j$. For $j > t$, $x_j = 0$.

Furthermore,

$$
q_t x_t + (1 - q_t)y_t = I_t(x, y) = I_t(x', y') = q_t x'_t
$$

$$
\iff q_t\left(1 - \sum_{j<t} x_j\right) + (1 - q_t)y_t = q_t x'_t
$$

$$
\iff q_t\left(1 - \sum_{j<t} x'_j\right) + (1 - q_t)y_t = q_t x'_t
$$

$$
\iff (1 - q_t)y_t = q_t\left(\sum_{j \leq t} x'_t - 1\right)
$$

Since $q_t, 1 - q_t > 0$, and $y_t \geq 0$, $\sum_{j \leq t} x'_t \leq 1$, it must be the case that $y_t = 0$ and $\sum_{j \leq t} x'_t = 1$. However, $y_t > 0$ by definition of $t$, and we have a contradiction.

This means that there is only one optimal basic feasible solution, and thus there is only one solution to the linear program. □

**Lemma 5.** *Let $0 < \alpha < 1$, $t := t(\alpha)$. Define*

$$
L_t = \sum_{j<t} \frac{1}{q_j}, \quad R_t = \sum_{j>t} \frac{1}{1 - q_j}.
$$

*Then*

$$
I^*_{\min} = \frac{1}{1 + q_t L_t + (1 - q_t)R_t},
$$

*and*

$$
x_j = \begin{cases} \frac{I^*_{\min}}{q_j}, & j < t \\ 1 - \frac{L_t}{1 + q_t L_t + (1 - q_t)R_t}, & j = t \\ 0, & j > t \end{cases}, \quad y_j = \begin{cases} 0, & j < t \\ 1 - \frac{R_t}{1 + q_t L_t + (1 - q_t)R_t}, & j = t \\ \frac{I^*_{\min}}{1 - q_j}, & j > t \end{cases}.
$$

*Proof.* Fix $0 < \alpha < 1$. This result mainly follows from the fact that $\sum_j x_j = \sum_j y_j = 1$, and $I_j(x^*, y^*) = I_{j'}(x^*, y^*)$ for all $j, j'$, as well as the sparse solution structure shown in Lemma 4.

We know that the optimal solution satisfies $I^*_{\min} = I_j = I_{j'}$ for all $j, j'$. We also already know that for $j > t$, $x_j = 0$, and for $j < t$, $y_j = 0$.

For $j < t$, $I^*_{\min} = I_j = x_j q_j + 0 \implies x_j = \frac{I^*_{\min}}{q_j}$.

For $j > t$, $I^*_{\min} = I_j = 0 + y_j(1 - q_j) \implies y_j = \frac{I^*_{\min}}{1-q_j}$.

For $j = t$, we know $I^*_{\min} = I_t = q_t x_t + (1 - q_t)y_t$. Then for $j < t$, $x_j = \frac{1}{q_j}(q_t x_t + (1 - q_t)y_t)$, so

$$1 - x_t = \sum_{j<t} x_j = \sum_{j<t} \frac{1}{q_j}(q_t x_t + (1-q_t)y_t) = (q_t x_t + (1-q_t)y_t)\sum_{j<t}\frac{1}{q_t} = (q_t x_t + (1-q_t)y_t)L_t. \tag{8}$$

Similarly, $y_j = \frac{1}{1-q_j}(q_t x_t + (1 - q_t)y_t)$ so

$$1 - y_t = \sum_{j>t} \frac{1}{1 - q_j}(q_t x_t + (1-q_t)y_t) = (q_t x_t + (1-q_t)y_t)R_t. \tag{9}$$

Combining equations 8 and 9, we obtain

$$\frac{L_t}{R_t}(1 - y_t) = 1 - x_t \implies x_t = 1 - \frac{L_t}{R_t}(1 - y_t) \tag{10}$$

Substituting equation 10 into equation 9, we get

$$1 - y_t = \left( q_t \left( 1 - \frac{L_t}{R_t}(1 - y_t)\right) + (1 - q_t)y_t \right) R_t$$
$$\iff 1 + q_t L_t - q_t R_t = y_t(1 + q_t L_t + (1 - q_t)R_t)$$
$$\implies y_t = \frac{1 + q_t L_t - q_t R_t}{1 + q_t L_t + (1 - q_t)R_t} = 1 - \frac{R_t}{1 + q_t L_t + (1 - q_t)R_t}$$

Then

$$x_t = 1 - \frac{L_t}{R_t}(1 - y_t) = 1 - \frac{L_t}{1 + q_t L_t + (1 - q_t)R_t}.$$

Finally, we see that

$$I^*_{\min} = q_t x_t + (1 - q_t)y_t$$
$$= q_t \left( 1 - \frac{L_t}{1 + q_t L_t + (1 - q_t)R_t}\right) + (1 - q_t)\left( 1 - \frac{R_t}{1 + q_t L_t + (1 - q_t)R_t}\right)$$
$$= 1 - \frac{q_t L_t + (1 - q_t)R_t}{1 + q_t L_t + (1 - q_t)R_t}$$
$$= \frac{1}{1 + q_t L_t + (1 - q_t)R_t}.$$

$\square$

**Lemma 6.** *For $0 < \alpha < 1/2$, if $\rho^*$ is the optimal recommendation policy, then $U_i(\rho^*, \alpha) \geq U_{i'}(\rho^*, \alpha)$ for $\tau(i) = 1$, $\tau(i') = 2$.*

*Proof.* Intuitively, the type with a larger proportion in the population must shoulder the burden of ensuring item fairness to mediocre items, and thus users from this type will have a lower recommendation probability on the items they really like compared to the smaller group.

Formally, let $i, i'$ be users such that $\tau(i) = 1, \tau(i') = 2$. For $\alpha \leq 1/2, j < t(\alpha)$,

$$
\begin{aligned}
x_j - y_{n-j+1} &= \frac{1}{q_j(\alpha)} - \frac{1}{1 - q_j(\alpha)} \\
&= \frac{\alpha v_j + (1-\alpha)v_{n-j+1}}{\alpha v_j} - \frac{\alpha v_{n-j+1} - (1-\alpha)v_j}{(1-\alpha)v_j} \\
&= \frac{1}{v_j}\left(v_j + \frac{1-\alpha}{\alpha}v_{n-j+1} - v_j - \frac{\alpha}{1-\alpha}v_{n-j+1}\right) \\
&= \frac{v_{n-j+1}}{v_j}\left(\frac{1-\alpha}{\alpha} - \frac{\alpha}{1-\alpha}\right) \\
&= \frac{v_{n-j+1}}{v_j}\frac{1-2\alpha}{\alpha(1-\alpha)} \\
&\geq 0 && (0 < \alpha \leq 1/2)
\end{aligned}
$$

So

$$
\begin{aligned}
\left(\max_j v_j\right) &\cdot (U_i - U_{i'}) \\
&= \sum_{j \leq t} v_j x_j - \sum_{j \geq t} v_{n-j+1} y_j \\
&= \sum_{j \leq t} v_j(x_j - y_{n-j+1}) - \sum_{j < t} v_j y_{n-j+1} && \text{(collecting coefficients of } v_j) \\
&\geq \sum_{j \leq t} v_t(x_j - y_{n-j+1}) - \sum_{j > t} v_j y_{n-j+1} && (v_t \leq v_j \text{ for } j \leq t, x_j > y_{n-j+1}) \\
&\geq v_t \sum_{j \leq t}(x_j - y_{n-j+1}) - v_t \sum_{j > t} y_{n-j+1} && (v_t > v_j \text{ for } j > t) \\
&\geq v_t - v_t \sum_{j \leq t} y_{n-j+1} - v_t \sum_{j > t} y_{n-j+1} && (\sum_{j \leq t} x_j = 1) \\
&= v_t - v_t && (\sum_{j \leq t} y_j = 1) \\
&= 0
\end{aligned}
$$

Since $\max_j v_j > 0$, we have that $U_i \geq U_{i'}$. $\qquad\square$

**Lemma 7.** $t(\alpha)$ *is weakly increasing.*

*Proof.* Suppose not, that is, there are some $\alpha < \alpha'$ such that $t := t(\alpha) > t(\alpha') := t'$. We will split this problem into several cases based on which of $\alpha, \alpha'$ gives a solution with higher item fairness, and show the implications of decreasing $t$ on the closed-form solutions from Lemma 5, eventually reaching a contradiction in each case.

Recall from Lemma 9 that $q_j(\alpha) = \frac{\alpha v_j}{\alpha v_j + (1-\alpha)v_{n-j+1}}$ is increasing in $\alpha$, so $q_j(\alpha) < q_j(\alpha')$ for all $j$.

First, let $I_{\min}^*(\alpha) < I_{\min}^*(\alpha')$. If $t = n$ then for $j < n$, $y_j = 0$, and thus $y_n = 1$. So

$$I_{\min}^*(\alpha) = (1 - q_n(\alpha)) > (1 - q_n(\alpha')) \geq (1 - q_n(\alpha'))y_n' = I_{\min}^*(\alpha')$$

since $y_n' \leq 1$, and we have a contradiction. Otherwise if $t < n$, let $j > t$. Then

$$(1 - q_j(\alpha))y_j = I_{\min}^*(\alpha) < I_{\min}^*(\alpha') = (1 - q_j(\alpha'))y_j' \iff \frac{y_j}{y_j'} < \frac{1 - q_j(\alpha')}{1 - q_j(\alpha)} < 1 \implies y_j < y_j'$$

Then $y_t = 1 - \sum_{j > t} y_j > 1 - \sum_{j > t} y_j' \geq y_t'$. So

$$
\begin{aligned}
q_t(\alpha)x_t + (1 - q_t(\alpha))y_t &< (1 - q_t(\alpha'))y_t' < (1 - q_t(\alpha'))y_t \\
\implies q_t(\alpha)x_t &< (1 - q_t(\alpha'))y_t - (1 - q_t(\alpha))y_t \\
\implies 0 \leq q_t(\alpha)x_t &< y_t(q_t(\alpha) - q_t(\alpha')) < 0
\end{aligned}
$$

and we have a contradiction.

Now, let $I^*_{\min}(\alpha) \geq I^*_{\min}(\alpha')$ instead. If $t' = 1$ then $x'_1 = 1$ and

$$I^*_{\min}(\alpha') = q_1(\alpha') > q_1(\alpha) \geq q_1(\alpha)x_1 = I^*_{\min}(\alpha),$$

and we have a contradiction. Otherwise if $t' > 1$, then for $j < t'$,

$$q_j(\alpha)x_j = I^*_{\min}(\alpha) \geq I^*_{\min}(\alpha') = q_j(\alpha')x'_j \implies \frac{x_j}{x'_j} \geq \frac{q_j(\alpha')}{q_j(\alpha)} > 1 \implies x_j > x'_j.$$

Then $x'_{t'} = 1 - \sum_{j<t'} x'_j > 1 - \sum_{j<t'} x_j \geq x_{t'}$. So

$$
\begin{aligned}
q_{t'}(\alpha')x'_{t'} + (1 - q_{t'}(\alpha'))y'_{t'} &= I^*_{\min}(\alpha') \leq I^*_{\min}(\alpha) = q_{t'}(\alpha)x_t \\
\implies (1 - q_{t'}(\alpha'))y'_{t'} &\leq q_{t'}(\alpha)x_{t'} - q_{t'}(\alpha')x'_{t'} < q_{t'}(\alpha)x_{t'} - q_{t'}(\alpha')x_{t'} \\
\implies 0 \leq (1 - q_{t'}(\alpha'))y'_{t'} &< x_{t'}(q_{t'}(\alpha) - q_{t'}(\alpha')) \leq 0
\end{aligned}
$$

and we have a contradiction. Thus all cases result in a contradiction. $\qquad\square$

**Lemma 8.** $I^*_{\min}(\alpha)$ is increasing for $\alpha \leq 1/2$.

*Proof.* We first need the following Lemma, which shows that $\alpha \leq 1/2$ implies that $t(\alpha)$ must be an index at most halfway through the list of items.

**Lemma 10.** If $\alpha \leq 1/2$, then $t(\alpha) \leq (n+1)/2$.

Now, for $1 \leq t \leq n$, define $\mathcal{A}(t) = \{\alpha : t(\alpha) = t\}$. Since $t(\alpha)$ is weakly increasing in $\alpha$ by Lemma 7, $\mathcal{A}(t)$ is an interval in $(0, 1)$, and $[\mathcal{A}(1), \mathcal{A}(2), ..., \mathcal{A}(n)]$ forms a consecutive partition of $(0, 1)$. We will now show that $I^*_{\min}(\alpha)$ is increasing on each of the first $\lfloor (n+1)/2 \rfloor$ intervals, which by Lemma 10 contain all $\alpha \in (0, 1/2]$.

**Lemma 11.** $I^*_{\min}(\alpha)$ is increasing on $\mathcal{A}(t)$ for $t \leq (n+1)/2$.

Finally, we show that this implies $I^*_{\min}(\alpha)$ is increasing on $0 < \alpha \leq 1/2$. Notice that $I_j(x, y, \alpha)$ is continuous in $\alpha$ for each fixed $x, y$, so $I_{\min}(x, y, \alpha) = \min_j I_j(x, y, \alpha)$ is continuous in $\alpha$ for each fixed $x, y$. Recall that $I^*_{\min}(\alpha) = \max_{x,y \in \Delta_{n-1}} I_{\min}(x, y, \alpha)$. As the supremum of a set of continuous functions on a compact set must itself be continuous , this means that $I^*_{\min}(\alpha)$ must be continuous in $\alpha$, since $\Delta_{n-1} \times \Delta_{n-1}$ is compact. Since $I^*_{\min}(\alpha)$ is continuous and is increasing on each interval $\{\mathcal{A}(t)\}_{t \leq (n+1)/2}$ by Lemma 11, $I^*_{\min}(\alpha)$ must be increasing on $\bigcup_{t \leq (n+1)/2} \mathcal{A}(t)$. By Lemma 10, $\bigcup_{t \leq (n+1)/2} \mathcal{A}(t) \supset (0, 1/2]$. So $I^*_{\min}(\alpha)$ is increasing on $(0, 1/2]$. $\qquad\square$

Now, we show Lemmas 10 and 11.

**Lemma 10.** If $\alpha \leq 1/2$, then $t(\alpha) \leq (n+1)/2$.

*Proof.* By Lemma 7, it is enough to show that for $\alpha = 1/2$, $t(\alpha) \leq (n+1)/2$.

Suppose that $\alpha = 1/2$ and we have $x, y$ of the form in Lemma 5 such that $t := t(\alpha) > (n+1)/2$.

If $n$ is even, take $t' = n/2$ in the definitions of $x_j, y_j$ in Lemma 5; if $n$ is odd, take $t' = (n+1)/2$. We now verify that this results in $x', y'$ that are a feasible solution to Problem 6 and that satisfy $x'_j = y'_{n-j+1}$. This is simple to see after observing that when $\alpha = 1/2$, $q_j = 1 - q_{n-j+1}$, so for even $n$, $L_{t'} = R_{t'} + \frac{1}{1-q_{t'}}$, and for odd $n$, $L_{t'} = R_{t'}$, and simplifying.

Since by assumption $t(\alpha)$ yields the optimal solution, $I_{\min}(x, y) > I_{\min}(x', y')$. By the construction of solutions in Lemma 5, for all $j$ we have $I_j(x', y') = I_{\min}(x', y')$ and $I_j(x, y) = I_{\min}(x, y)$. Then

for $j < t$, $q_j x_j = I_j(x, y) > I_j(x', y') = q_j x_j'$, so $x_j > x_j'$. Then since $I_{t'}(x, y) > I_{t'}(x', y')$,

$$q_{t'} x_{t'} > q_{t'} x_{t'}' + (1 - q_{t'}) y_{t'}'$$

$$= q_{t'} \left( 1 - \sum_{j < t'} x_j' \right) + (1 - q_{t'}) y_{t'}'$$

$$> q_{t'} \left( 1 - \sum_{j < t'} x_j \right) + (1 - q_{t'}) y_{t'}'$$

$$> q_{t'} \left( 1 - \sum_{j < t'} x_j \right)$$

$$\geq q_{t'} x_{t'}$$

which is a contradiction. $\qquad \square$

**Lemma 11.** $I_{\min}^*(\alpha)$ *is increasing on* $\mathcal{A}(t)$ *for* $t \leq (n + 1)/2$.

*Proof.* Recall that for fixed $t$,

$$I_{\min}^*(\alpha) = \frac{1}{1 + q_t(\alpha) L_t(\alpha) + (1 - q_t(\alpha)) R_t(\alpha)}.$$

It is sufficient to show that $q_t L_t + (1 - q_t) R_t$ is decreasing.

We first expand the expression:

$$q_t(\alpha) L_t(\alpha) + (1 - q_t(\alpha)) R_t(\alpha)$$

$$= \sum_{j < t} \frac{q_t(\alpha)}{q_j(\alpha)} + \sum_{j > t} \frac{1 - q_t(\alpha)}{1 - q_j(\alpha)}$$

$$= \sum_{j < t} \frac{q_t(\alpha)}{q_j(\alpha)} + \sum_{j \geq n-t} \frac{1 - q_t(\alpha)}{1 - q_j(\alpha)} + \sum_{t < j < n-t} \frac{1 - q_t(\alpha)}{1 - q_j(\alpha)}$$

Let $t < j \leq n - t + 1$. Then

$$g(\alpha) := \frac{1 - q_t(\alpha)}{1 - q_j(\alpha)}$$

$$= \frac{(1 - \alpha) v_{n-t+1}}{\alpha v_t + (1 - \alpha) v_{n-t+1}} \frac{\alpha v_j + (1 - \alpha) v_{n-j+1}}{(1 - \alpha) v_{n-j+1}}$$

$$= \frac{v_{n-t+1}}{v_{n-j+1}} \frac{\alpha v_j + (1 - \alpha) v_{n-j+1}}{\alpha v_t + (1 - \alpha) v_{n-t+1}}$$

$$= \frac{v_{n-t+1}}{v_{n-j+1}} \frac{v_{n-j+1} + \alpha (v_j - v_{n-j+1})}{v_{n-t+1} + \alpha (v_t - v_{n-t+1})}$$

So

$$g'(\alpha) = \frac{v_{n-t+1}}{v_{n-j+1}} \frac{(v_j - v_{n-j+1})(v_{n-t+1} + \alpha(v_t - \alpha) v_{n-t+1})) - (v_t - v_{n-t+1})(v_{n-j+1} + \alpha(v_j - v_{n-j+1}))}{(v_{n-t+1} + \alpha(v_t - v_{n-t+1}))^2}$$

$$\propto (v_j - v_{n-j+1})(v_{n-t+1} + \alpha(v_t - v_{n-t+1})) - (v_t - v_{n-t+1})(v_{n-j+1} + \alpha(v_j - v_{n-j+1}))$$

$$= (v_j - v_{n-j+1}) v_{n-t+1} - (v_t - v_{n-t+1}) v_{n-j+1}$$

$$= v_j v_{n-t+1} - v_t v_{n-j+1}$$

$$\leq v_t v_{n-j+1} - v_t v_{n-j+1} \qquad\qquad (j > t, n - j + 1 < n - t + 1)$$

$$= 0$$

So $g$ is a decreasing function of $\alpha$. Now, consider

$$\sum_{j<t} \frac{q_t(\alpha)}{q_j(\alpha)} + \sum_{j>n-t} \frac{1-q_t(\alpha)}{1-q_j(\alpha)}$$

$$= \sum_{j=1}^{t-1} \frac{q_t(\alpha)}{q_j(\alpha)} + \sum_{j=n-t+2}^{n} \frac{1-q_t(\alpha)}{1-q_j(\alpha)}$$

$$= \sum_{j=1}^{t-1} \frac{q_t(\alpha)}{q_j(\alpha)} + \sum_{i=1}^{t-1} \frac{1-q_t(\alpha)}{1-q_{n-i+1}(\alpha)} \qquad (i = n-j+1)$$

$$= \sum_{j=1}^{t-1} \frac{q_t(\alpha)}{q_j(\alpha)} + \frac{1-q_t(\alpha)}{1-q_{n-j+1}(\alpha)}$$

Then for $1 \le j \le t-1$,

$$h_t(\alpha) := \frac{q_t(\alpha)}{q_j(\alpha)} + \frac{1-q_t(\alpha)}{1-q_{n-j+1}(\alpha)}$$

$$= \frac{\alpha v_t}{\alpha v_t + (1-\alpha)v_{n-t+1}} \cdot \frac{\alpha v_j + (1-\alpha)v_{n-j+1}}{\alpha v_j}$$

$$+ \frac{(1-\alpha)v_{n-t+1}}{\alpha v_t + (1-\alpha)v_{n-t+1}} \cdot \frac{\alpha v_{n-j+1} + (1-\alpha)v_j}{(1-\alpha)v_j}$$

$$= \frac{v_t(\alpha v_j + (1-\alpha)v_{n-j+1}) + v_{n-t+1}(\alpha v_{n-j+1} + (1-\alpha)v_j)}{v_j(\alpha v_t + (1-\alpha)v_{n-t+1})}$$

$$= 1 + \frac{(1-\alpha)v_t v_{n-j+1} + \alpha v_{n-t+1}v_{n-j+1}}{v_j(\alpha v_t + (1-\alpha)v_{n-t+1})}$$

$$= 1 + \frac{v_{n-j+1}}{v_j} \cdot \frac{(1-\alpha)v_t + \alpha v_{n-t+1}}{\alpha v_t + (1-\alpha)v_{n-t+1}}$$

$$= 1 + \frac{v_{n-j+1}}{v_j} \cdot \frac{v_t + \alpha(v_{n-t+1} - v_t)}{v_{n-t+1} + \alpha(v_t - v_{n-t+1})}.$$

So

$$h_t'(\alpha) = \frac{v_{n-j+1}}{v_j}$$

$$\cdot \frac{(v_{n-t+1} - v_t)(v_{n-t+1} + \alpha(v_t - v_{n-t+1})) - (v_t - v_{n-t+1})(v_t + \alpha(v_{n-t+1} - v_t))}{(v_{n-t+1} + \alpha(v_t - v_{n-t+1}))^2}$$

$$\propto -(v_t - v_{n-t+1})(v_{n-t+1} + \alpha(v_t - v_{n-t+1})) - (v_t - v_{n-t+1})(v_t + \alpha(v_{n-t+1} - v_t))$$

$$= -(v_t - v_{n-t+1})(v_{n-t+1} + \alpha(v_t - v_{n-t+1}) + v_t + \alpha(v_{n-t+1} - v_t))$$

$$= -(v_t - v_{n-t+1})(v_t + v_{n-t+1})$$

$$\le -(v_t - v_{n-t+1}) \qquad (t \le (n+1)/2 \le n-t+1)$$

$$\le 0$$

and $h_t$ is decreasing in $\alpha$ for $1 \le j \le t-1$. Then $\sum_{j=1}^{t-1} h_t(\alpha)$ is decreasing, and $q_t L_t + (1-q_t)R_t = \sum_{j=1}^{t-1} h_t(\alpha) + \sum_{j=t+1}^{n-t+1} g_t(\alpha)$ is decreasing, and $I_{\min}^*(\alpha)$ is increasing. $\qquad\square$

## E  Proof of Theorem 4

Recall Theorem 4.

**Theorem 4.** *If $\beta > \frac{1}{n}$ and $w$ and $\hat{w}$ are as described above, then fairness constraints can arbitrarily worsen the price of misestimation.*

- *The price of misestimation without fairness constraints is low: for all $\{v_j\}$, there is a recommendation policy $\hat{\rho}$ that solves the misestimated problem $U_{\min}^*(0, \hat{w})$ so that*

$$\pi_U^M(0, w, \hat{w}) \le \frac{1}{2}.$$

- *The price of misestimation with fairness constraints can be arbitrarily large: $\forall \epsilon$, there exists $\{v_j\}$ and a recommendation policy $\hat{\rho}$ that solves the problem $U^*_{\min}(1, \hat{w})$ such that*

$$\pi^M_U(1, w, \hat{w}) > 1 - \epsilon.$$

## E.1 Main proof

*Proof.* We first find a worst-case price of misestimation with fairness constraints, and then find the price of misestimation without fairness constraints.

**Item fairness constraints with misestimation.** First, notice that in the misestimated value matrix $\hat{w}$ there are three distinct user value types, and thus we can identify $\mathcal{S}_{\text{symm}}$ with $\Delta^3_{n-1}$ and a policy $\rho \in \mathcal{S}_{\text{symm}}$ with vectors $x, y, z$, where $x$ is the recommendation policy for users with value $\hat{w}_{ij} = v_j$, $y$ is the recommendation policy for users with value $\hat{w}_{ij} = v_{n-j+1}$, and $z$ is the recommendation policy for the mis-estimated users.

Let
$$\mathcal{S}' = \{x, y, z \in \mathcal{S}_{\text{symm}} : x_j = y_{n-j+1}, z_j = z_{n-j+1} \text{ for all } j\}.$$
Since $y$ is completely determined by $x$, we can identify $\mathcal{S}'$ with $\Delta^2_{n-1}$ and can identify $x, y, z \in \mathcal{S}'$ simply by $x, z$.

**Lemma 12.** *If a policy $\rho = (x, y, z) \in \mathcal{S}_{\text{symm}}$ solves $U_{\min}(\rho) = U^*_{\min}(1)$, then there is some policy $\rho' = x', z' \in \mathcal{S}'$ that solves $U_{\min}(\rho') = U^*_{\min}(1)$.*

Applying the framework in Proposition 1 to this proof with $\mathcal{S} = \mathcal{S}'$ and using $\hat{w}$ as the user and item values, we reduce the problem to finding

$$\phi = \arg \max_{x, z \in \Delta^2_{n-1}, \lambda} \lambda$$
$$\text{subject to} \quad I_j(x, z) = \lambda \quad \forall j,$$
(11)

and showing that this $\phi$ is unique.

First, we use an analogue of the sparsity result Proposition 2 to show that the optimal $x, z$ has the following sparse structure.

**Lemma 13.** *Let $x, z$ be an optimal basic feasible solution to the linear program Problem 11. There is some $j \leq \frac{n+1}{2}$ such that for all $j' > j$, $x_{j'} = 0$ and for all $j' < j$, $z_{j'} = z_{n-j'+1} = 0$. Let the pivot index $t$ denote the minimum such $j$.*

This sparse structure implies uniqueness.

**Lemma 14.** *Problem 11 has a unique optimal solution.*

We can now describe what this solution $(x, z)$ looks like.

**Lemma 15.** *Let $(x, z)$ be the optimal solution to Problem 12, and let $t$ be the pivot element of $(x, z)$; suppose that $t \neq \frac{n+1}{2}$. Define*
$$L_t := \sum_{j < t} \frac{1}{q_j}.$$
*Then*
$$\lambda = \frac{2\beta q_t + \frac{1}{2}(1 - 2\beta)}{1 + q_t L_t + \frac{1}{2}(n - 2t)},$$
$$z_j = \begin{cases} 0, & j < t \\ \frac{1}{2}\left(1 - (n - 2t)\frac{\lambda}{1 - 2\beta}\right), & j \in \{t, n - t + 1\} \\ \frac{\lambda}{1 - 2\beta}, & t < j < n - t + 1 \end{cases}.$$
*Finally,*
$$x_j = \begin{cases} \frac{\lambda}{2\beta q_j}, & j < t \\ 1 - \frac{\lambda}{2\beta} L_t, & j = t \\ 0, & j > t \end{cases}.$$

*If $t = \frac{n+1}{2}$, then $x$ remains the same, but $z_t = 1$ and*

$$\lambda = \frac{2\beta q_t + (1 - 2\beta)}{1 + q_t L_t}.$$

If we try $t = 1$ and obtain $z_1 < 0$ above, we can conclude that $t > 1$. Suppose $t = 1$. Then $L_t = 0$, so

$$\lambda = \frac{2\beta q_1 + \frac{1}{2}(1 - 2\beta)}{1 + \frac{1}{2}(n - 2)}$$

and

$$z_1 = \frac{1}{2}\left(1 - (n-2)\frac{\lambda}{1 - 2\beta}\right) = \frac{1}{2}\left(1 - (n-2)\frac{1}{1 - 2\beta}\frac{2\beta q_1 + \frac{1}{2}(1 - 2\beta)}{n/2}\right) = \frac{1}{n} - \frac{n-2}{n}\frac{2\beta q_1}{1 - 2\beta}$$

Then $t = 1$ and $z_1 < 0$ if and only if

$$\frac{1}{n} - \frac{n-2}{n}\frac{2\beta q_1}{1 - 2\beta} < 0 \iff \frac{v_n}{v_1} < \frac{n-2}{\frac{1}{2\beta} - 1} - 1.$$

since $q_1 = \frac{v_1}{v_1 + v_n}$. If $\beta > \frac{1}{n}$, then

$$\frac{n-2}{\frac{1}{2\beta} - 1} - 1 > \frac{n-2}{n/2 - 1} - 1 = 2\frac{n-2}{n-2} - 1 = 1.$$

So if $\beta > \frac{1}{n}$, $t > 1$ and $z_1 = z_n = 0$. In this case, regardless of whether a mis-estimated user has $w_{i1} = v_1$ or $w_{in} = v_1$, that user will *never* be recommended their most preferred item.

Moreover, no matter the true type of the mis-estimated user,

$$U_i(x, z) = \frac{1}{v_1}\left(\sum_{1 \le j \le n} v_j z_j\right) = \frac{1}{v_1}\left(\sum_{1 < j < n} v_j z_j\right) < \frac{v_2}{v_1}\left(\sum_{1 < j < n} z_j\right) = \frac{v_2}{v_1}$$

Thus if $v_2 < \frac{v_1}{n}\epsilon$, then $U_i(x, z) < \frac{\epsilon}{n}$.

Taking $\hat\rho$ to be the recommendation probabilities induced by this pair $x, z$, this implies that $U_{\min}(\hat\rho) < \epsilon/n$.

**Item fairness constraints without mis-estimation.** This becomes precisely the problem we solve in Theorem 3; if the two types occur in equal proportion in the mis-estimated group as well as the correctly estimated group, then $\alpha = 1/2$.

If $\alpha = 1/2$, then by Lemma 17 with $\beta = 1$, $t = \lfloor\frac{n+1}{2}\rfloor$, and we can show that

$$L_t = \begin{cases} R_t - \frac{1}{q_t}, & n \text{ even} \\ R_t, & n \text{ odd} \end{cases}.$$

By Lemma 5 we know that

$$I^*_{\min} = \frac{1}{1 + q_t L_t + (1 - q_t)R_t},$$

so if $n$ is even, we have

$$I^*_{\min} = \frac{1}{1 + q_t(R_t - \frac{1}{q_t}) + (1 - q_t)R_t} = \frac{1}{R_t} = \frac{1}{\sum_{j>t}\frac{1}{1-q_j}} > \frac{1}{2(n - (n/2))} = \frac{1}{n}$$

where we use the fact from Lemma 16 that if $j > \frac{n+1}{2}$, then $q_j < \frac{1}{2}$. Likewise if $n$ is odd,

$$I^*_{\min} = \frac{1}{1 + q_t R_t + (1 - q_t)R_t} = \frac{1}{1 + R_t} = \frac{1}{1 + \sum_{j>t}\frac{1}{1-q_j}} > \frac{1}{1 + 2(n - (n+1)/2)} = \frac{1}{n}.$$

Then a user $i$ of type 1 will have normalized utility will be

$$U_i(x, y) \ge \frac{x_1 v_1}{v_1} = \frac{I^*_{\min}}{q_1} = I^*_{\min}\frac{v_1 + v_n}{v_n} \ge I^*_{\min} > \frac{1}{n}.$$

Similarly a user of type 2 will have utility

$$U_i(x, y) \ge \frac{y_n v_1}{v_1} = \frac{I^*_{\min}}{1 - q_n} = I^*_{\min}\frac{v_1 + v_n}{v_n} \ge I^*_{\min} > \frac{1}{n}.$$

This means that $U^*_{\min}(1) > 1/n$.

**Price of estimation with item fairness constraints.** The two arguments above imply that

$$\pi_U^M(1, w, \hat{w}) = 1 - \frac{U_{\min}(\hat{\rho}, w)}{U_{\min}^*(1, w)}$$

$$> 1 - \frac{\epsilon/n}{1/n}$$

$$= 1 - \epsilon.$$

**Price of estimation without item fairness constraints.** In this case an optimal recommendation policy may choose each user's policy individually without regard to item utilities. Given the mis-estimated values, it is optimal to give mis-estimated users the item $j^*$ or $n - j^* + 1$ that maximizes $\frac{v_j + v_{n-j+1}}{2}$ deterministically. It is thus also optimal to recommend the item $j^*$ and $n - j^* + 1$ each with probability $1/2$. This yields expected value

$$\max_j \frac{v_j + v_{n-j+1}}{2} \geq \frac{v_1 + v_n}{2} > \frac{v_1}{2},$$

and thus a mis-estimated user $i$ will receive a normalized value of $U_i(\rho') = 1/2$.

The optimal utility of each user if their preferences were correctly estimated is 1, so the price of misestimation is $1/2$. □

### E.2 Supporting lemma proofs

First, we show a fact that will be helpful later on.

**Lemma 16.** *If $j < \frac{n+1}{2}$, then $q_j > 1/2$; if $j > \frac{n+1}{2}$, then $q_j < 1/2$. If $j = \frac{n+1}{2}, q_j = 1/2$.*

*Proof.* If $j < \frac{n+1}{2}$, then $v_j > v_{n-j+1}$ and

$$q_j = \frac{v_j}{v_j + v_{n-j+1}} = \frac{1}{1 + \frac{v_{n-j+1}}{v_j}} > \frac{1}{1+1} = \frac{1}{2};$$

if $j < \frac{n+1}{2}$, then $v_j < v_{n-j+1}$ and a symmetric argument holds.

If $j = \frac{n+1}{2}$, then $v_j = v_{n-j+1}$ and thus $q_j = 1/2$. □

In the remainder of this section, we will use $\mathrm{rev}(x)$ to denote the reverse of a vector $x$, that is, $\mathrm{rev}(x)_j = x_{n-j+1}$ for all $j$.

**Lemma 12.** *If a policy $\rho = (x, y, z) \in \mathcal{S}_{\mathrm{symm}}$ solves $U_{\min}(\rho) = U_{\min}^*(1)$, then there is some policy $\rho' = x', z' \in \mathcal{S}'$ that solves $U_{\min}(\rho') = U_{\min}^*(1)$.*

*Proof.* Suppose $(x, y, z) := \rho$ is an optimal solution to $U_{\min}^*(1)$.

First, if $U_{\min}^*(1) = U_i(\rho)$ for some $i$ in the first or second type of user

Let $\rho' = x', y', z'$ where $x'_j = y'_{n-j+1} = \frac{1}{2}(x_j + y_{n-j+1})$, and $z'_j = \frac{1}{2}(z_j + z_{n-j+1})$.

Then clearly $\rho' \in \mathcal{S}'$, and $\sum_j x'_j = \sum_j y'_j = \sum_j z'_j = 1, 0 \leq x'_j, y'_j, z'_j \leq 1$ for all $j$.

Furthermore, for users $i, i'$ of the first and second types respectively with correctly estimated types,

$$U_i(\rho') = \sum_{j=1}^n x'_j v_j = \sum_{j=1}^n \frac{1}{2}(x_j + y_{n-j+1})v_j = \frac{U_i(\rho) + U_{i'}(\rho)}{2} \geq \min\{U_i(\rho), U_{i'}(\rho)\}$$

and by an analogous argument,

$$U_{i'} \geq \min\{U_i(\rho), U_{i'}(\rho)\}.$$

Moreover, the estimated normalized utility of a mis-estimated user $i$, will be

$$U_i(\rho') = \sum_{j=1}^{n} z_j' \frac{v_j + v_{n-j+1}}{2}$$

$$= \frac{1}{2} \sum_{j=1}^{n} z_j \frac{v_j + v_{n-j+1}}{2} + \frac{1}{2} \sum_{j=1}^{n} z_{n-j+1} \frac{v_j + v_{n-j+1}}{2}$$

$$= \frac{1}{2} (U_i(\rho') + U_i(\rho'))$$

$$= U_i(\rho').$$

So $\min_i U_i(\rho') \geq \min_i U_i(\rho)$.

Finally, for any $j$, we can expand the definition of $I_j$ to see that

$$I_j(\rho') = \frac{1}{2}(I_j(\rho) + I_{n-j+1}(\rho)) = I_{\min}^*.$$

Thus $\rho'$ is also an optimal solution to Problem 1. $\qquad\square$

**Lemma 13.** *Let $x, z$ be an optimal basic feasible solution to the linear program Problem 11. There is some $j \leq \frac{n+1}{2}$ such that for all $j' > j$, $x_{j'} = 0$ and for all $j' < j$, $z_{j'} = z_{n-j'+1} = 0$. Let the pivot index $t$ denote the minimum such $j$.*

*Proof.* **Part 1.** While it is tempting attempt to directly apply the sparsity result in Proposition 2 here, we will need a slightly stronger result to ensure that there are no $j$ such that $x_j = z_j = 0$. First, we must show that $x_j > 0$ and $y_j > 0$ are almost mutually exclusive.

**Lemma 17.** *Let $(x, z)$ be an optimal solution to Problem 11. For $j > \frac{n+1}{2}$, $x_j = 0$.*

In particular, if we let $h = \lfloor \frac{n+1}{2} \rfloor$ indicate the index halfway through the set of items, this means that we can simplify the problem above to

$$\arg \max_{\substack{x_1, \ldots, x_h, \\ z_1, \ldots, z_h, \\ \lambda}} \quad \lambda$$

$$\text{subject to} \quad I_j(x, z) = \lambda, 1 \leq j \leq h$$

$$\sum_{j \leq h} x_j = \sum_{j \leq h} z_j + \sum_{h < j \leq n} z_{n-j+1} = 1 \qquad (12)$$

$$x_j, z_j \geq 0.$$

Now this is a linear program with $2h+1$ variables and $h+2$ constraints, so by an argument analogous to Problem 12, we find that every basic feasible solution $x_{1:h}, z_{1:h}$ must share $h-1$ zeros. This means that there is a single index $t$ such that $x_t > 0, z_t > 0$; if there were more than one, by the pigeonhole principle there must be some $t' \leq h$ such that $x_{t'} = z_{t'} = x_{n-t'+1} = 0$, and $I_{t'}(x, z) = 0$. Then $I_{\min}(x, z) = 0$; however, by Lemma 1, $I_{\min}(x, z) = I_{\min}^* > 0$, so we have a contradiction.

In terms of $x, z$,

$$I_j(x, z) = \frac{\beta v_j x_j + \beta v_{n-j+1} x_{n-j+1} + (1 - 2\beta) \frac{v_j + v_{n-j+1}}{2} z_j}{\beta v_j + \beta v_{n-j+1} + (1 - 2\beta) \frac{v_j + v_{n-j+1}}{2}}$$

$$= \frac{\beta v_j x_j + \beta v_{n-j+1} x_{n-j+1} + (1 - 2\beta) \frac{v_j + v_{n-j+1}}{2} z_j}{\frac{v_j + v_{n-j+1}}{2}}$$

$$= 2\beta \left( \frac{v_j}{v_j + v_{n-j+1}} x_j + \frac{v_{n-j+1}}{v_j + v_{n-j+1}} x_{n-j+1} \right) + (1 - 2\beta) z_j$$

Let $q_j = \frac{v_j}{v_j + v_{n-j+1}}$. Note that $q_j$ does not depend on $\beta$, and $q_j$ is decreasing in $j$. Then

$$I_j(x, y, z) = 2\beta \left( q_j x_j + (1 - q_j) y_j \right) + (1 - 2\beta) z_j.$$

**Part 2.** Moreover, the set of $j$ such that $x_j = 0$ is contiguous; similarly for $z_j$.

We prove this by contradiction, showing that if this set is not contiguous, we can move around probability mass until we achieve greater than optimal item fairness. Let $i < j \leq h$, and let $x, z$ be an optimal solution to Problem 12. We need to show that if $x_i = 0$ then $x_j = 0$, and if $z_i > 0$, then $z_j > 0$.

Suppose first that $x_i = 0$ and $x_j > 0$. Notice that since $x, z$ are optimal,

$$(1 - 2\beta) z_i = I_i(x, z) = I_j(x, z) = 2\beta q_j x_j + (1 - 2\beta) z_j$$

Define

$$x'_\ell = \begin{cases} x_j - \epsilon, & \ell = i \\ x_i + \frac{\epsilon}{h-1}, & \ell = j \\ x_\ell + \frac{\epsilon}{h-1}, & \text{otherwise} \end{cases} \quad , \quad z'_\ell = \begin{cases} z_j, & \ell = i \\ z_i, & \ell = j \\ z_\ell, & \text{otherwise} \end{cases}$$

for $\epsilon$ small enough that $x' \in \Delta_{h-1}$. Now for all $\ell \notin \{i, j\}$, $I_\ell(x', z') > I_\ell(x, z)$, since we have increased the probability mass on $x_\ell$ while leaving $z_\ell$ unchanged. Furthermore,

$$\begin{aligned} I_i(x', z') &= 2\beta q_i (x_j - \epsilon) + (1 - 2\beta) z_j \\ &= 2\beta (q_i - q_j) x_j + 2\beta q_j x_j + (1 - 2\beta) z_j - 2\beta q_i \epsilon \\ &= 2\beta (q_i - q_j) x_j + (1 - 2\beta) z_i - 2\beta q_i \epsilon \\ &= I_i(x, z) + 2\beta (q_i - q_j) x_j - 2\beta q_i \epsilon \\ &> I_i(x, z) \end{aligned}$$

as long as $q_i \epsilon < (q_i - q_j) x_j$, which we can ensure by taking $\epsilon$ small enough, since $q_i > q_j$. Moreover,

$$\begin{aligned} I_j(x', z') &= 2\beta q_j \left( x_i + \frac{\epsilon}{h-1} \right) + (1 - 2\beta) z_i \\ &= 2\beta q_j \left( x_i + \frac{\epsilon}{h-1} \right) + 2\beta q_j x_j + (1 - 2\beta) z_j \\ &> 2\beta q_j x_j + (1 - 2\beta) z_j \\ &= I_j(x, z). \end{aligned}$$

Thus

$$I_{\min}(x', z') = \min_\ell I_\ell(x', z') > \min_\ell I_\ell(x, z) = I_{\min}(x, z) = I_{\min}^*$$

and we have a contradiction.

Similarly, suppose that $z_i > 0$ and $z_j = 0$. Then since $x, z$ is optimal, we have

$$2\beta q_i x_i + (1 - 2\beta) z_i = I_i(x, z) = I_j(x, z) = 2\beta q_j x_j. \tag{13}$$

Define

$$z'_\ell = \begin{cases} z_i - \epsilon, & \ell = j \\ z_j + \frac{\epsilon}{h-1}, & \ell = i \\ z_\ell + \frac{\epsilon}{h-1}, & \text{otherwise} \end{cases} \quad , \quad x'_\ell = \begin{cases} x_j - \delta, & \ell = j \\ x_i + \delta, & \ell = i \\ x_\ell, & \text{otherwise} \end{cases}$$

for $\delta = \frac{1-2\beta}{2\beta q_i} z_i > 0$ and taking $\epsilon$ small enough that $z' \in \Delta_{h-1}$ and $\epsilon < z_i(1 - q_j/q_i)$. This implies that

$$\delta = \frac{1 - 2\beta}{2\beta q_i} z_i < \frac{(1 - 2\beta)(z_i - \epsilon)}{2\beta q_j}$$

Now

$$x_i + \delta = x_i + \frac{1 - 2\beta}{2\beta q_i} z_i$$

$$= \frac{1}{2\beta q_i} (2\beta q_i x_i + (1 - 2\beta) z_i)$$

$$= \frac{1}{2\beta q_i} 2\beta q_j x_j \qquad (13)$$

$$= \frac{q_j}{q_i} x_j$$

$$< x_j \qquad (q_j < q_i)$$

$$\leq 1$$

and

$$x_j - \delta > x_j - \frac{(1 - 2\beta)(z_i - \epsilon)}{2\beta q_j}$$

$$= \frac{1}{2\beta q_j} (2\beta q_j x_j - (1 - 2\beta)(z_i - \epsilon))$$

$$> \frac{1}{2\beta q_j} (2\beta q_j x_j - (1 - 2\beta) z_i)$$

$$= \frac{1}{2\beta q_j} (2\beta q_j x_j - (2\beta q_j x_j - 2\beta q_i x_i)) \qquad (13)$$

$$= \frac{1}{2\beta q_j} 2\beta q_i x_i$$

$$> 0$$

and $x_i + \delta > x_i \geq 0$, $x_j - \delta < x_j \leq 1$, so $x' \in \Delta_{h-1}$.

Now as above $I_\ell(x', z') > I_{\min}(x, z)$ for all $\ell \notin \{i, j\}$, since there is strictly more probability mass on $z_\ell$ while $x_\ell$ remains the same. Furthermore,

$$I_i(x', z') - I_i(x, z) = 2\beta q_i(x_i + \delta) + (1 - 2\beta)(z_j + \frac{\epsilon}{h - 1}) - (2\beta q_i x_i + (1 - 2\beta) z_i)$$

$$> 2\beta q_i(x_i + \delta) - (2\beta q_i x_i + (1 - 2\beta) z_i)$$

$$= 2\beta q_i \delta - (1 - 2\beta) z_i$$

$$= 2\beta q_i \frac{(1 - 2\delta)}{2\beta q_i} z_i - (1 - 2\beta) z_i$$

$$= 0$$

and

$$I_j(x', z') - I_j(x, z) = 2\beta q_j(x_j - \delta) + (1 - 2\beta)(z_i - \epsilon) - 2\beta q_j x_j$$

$$= (1 - 2\beta)(z_i - \epsilon) - 2\beta q_j \delta$$

$$> (1 - 2\beta)(z_i - \epsilon) - 2\beta q_j \frac{(1 - 2\beta)(z_i - \epsilon)}{2\beta q_j}$$

$$= 0$$

Then

$$I_{\min}(x', z') = \min_\ell I_\ell(x', z') > \min_\ell I_\ell(x, z) = I_{\min}(x, z) = I^*_{\min}$$

and we have found $x', z'$ achieving a higher-than-optimal $I_{\min}$, which is a contradiction. $\qquad \square$

**Lemma 17.** *Let $(x, z)$ be an optimal solution to Problem 11. For $j > \frac{n+1}{2}$, $x_j = 0$.*

*Proof.* We prove this by contradiction. Suppose that for $j > \frac{n+1}{2}$, $x_j > 0$. Then define $x'$ such that

$$x'_\ell = \begin{cases} 0, & \ell = j \\ x_{n-j+1} + (x_j - \epsilon) & \ell = n - j + 1 \\ x_\ell + \frac{\epsilon}{n-2} \end{cases}$$

where $0 < \epsilon < (2q_{n-j+1} - 1)x_j$. Since $j > \frac{n+1}{2}$, $q_{n-j+1} > 1/2$ by Lemma 16, so such $\epsilon$ exists. Then for all $\ell \notin \{j, n-j+1\}$,

$$
\begin{aligned}
I_\ell(x', z) &= 2\beta \left( q_\ell x'_\ell + (1 - q_\ell) x'_{n-\ell+1} \right) + (1 - 2\beta)z_\ell \\
&> 2\beta \left( q_\ell x_\ell + (1 - q_\ell) x_{n-\ell+1} \right) + (1 - 2\beta)z_\ell \\
&= I_\ell(x, z).
\end{aligned}
$$

Since $\epsilon < (2q_{n-j+1} - 1)x_j$, we can rearrange to see that $(1 - q_{n-j+1})x_j < q_{n-j+1}(x_j - \epsilon)$. This implies that

$$
\begin{aligned}
I_{n-j+1}(x', z) &= 2\beta \left( q_{n-j+1}(x_{n-j+1} + (x_j - \epsilon)) + q_j \cdot 0 \right) + (1 - 2\beta)z_{n-j+1} \\
&= 2\beta \left( q_{n-j+1}x_{n-j+1} + q_{n-j+1}(x_j - \epsilon) \right) + (1 - 2\beta)z_{n-j+1} \\
&> 2\beta \left( q_{n-j+1}x_{n-j+1} + 2\beta(1 - q_{n-j+1})x_j \right) + (1 - 2\beta)z_{n-j+1} \\
&= I_{n-j+1}(x, z)
\end{aligned}
$$

and

$$
\begin{aligned}
I_j(x', z) &= 2\beta \left( q_j \cdot 0 + (1 - q_j) \cdot (x_{n-j+1} + (x_j - \epsilon)) \right) + (1 - 2\beta)z_j \\
&\quad 2\beta \left( q_{n-j+1}x_{n-j+1} + q_{n-j+1}(x_j - \epsilon) \right) + (1 - 2\beta)z_j & (q_{n-j+1} = 1 - q_j) \\
&> 2\beta \left( q_{n-j+1}x_{n-j+1} + (1 - q_{n-j+1})x_j \right) + (1 - 2\beta)z_j \\
&= 2\beta \left( (1 - q_j)x_{n-j+1} + q_j x_j \right) + (1 - 2\beta)z_j & (q_{n-j+1} = 1 - q_j) \\
&= I_j(x, z)
\end{aligned}
$$

Thus we have found a solution $x', z$ that satisfies

$$
I_{\min}(x', z) = \min_j I_j(x', z) > \min_j I_j(x, z) = I_{\min}(x, z) = I^*_{\min}
$$

and we have a contradiction. $\qquad\square$

**Lemma 14.** *Problem 11 has a unique optimal solution.*

*Proof.* Suppose $(x, z)$ and $(x', z')$ are both optimal basic feasible solutions to Problem 12. Let $t$ be the pivot index of $(x, z)$ and $t'$ be the pivot index of $(x', z')$, and suppose without loss of generality that $t \leq t'$.

First, if $t < t'$, then for all $j < t$, $z_j = z'_j = 0$, and

$$
2\beta q_j x_j = I_j(x, z) = I_{\min}(x, z) = I_{\min}(x', z') = I_j(x', z') = 2\beta q_j x'_j
$$

so $x_j = x'_j$. If $j > t$, $x_j = 0$. For $j = t$, $z'_t = 0$ and

$$
2\beta q_t x_t + (1 - 2\beta)z_t = 2\beta q_t x'_t
$$

$$
\implies 2\beta q_t \left( 1 - \sum_{j<t} x_j \right) + (1 - 2\beta)z_t = 2\beta q_t x'_t
$$

$$
\implies 2\beta q_t \left( 1 - \sum_{j<t} x'_j \right) + (1 - 2\beta)z_t = 2\beta q_t x'_t
$$

$$
\implies (1 - 2\beta)z_t = 2\beta q_t \left( \sum_{j\leq t} x'_j - 1 \right).
$$

But $(1 - 2\beta)z_t \geq 0$, so $\sum_{j\leq t} x'_j = 1$ and for $j > t$, $x'_j = 0$. This implies that $t' \leq t$, which is a contradiction, so it is not possible for $t' > t$.

Now, if $t = t'$, then again for all $j < t$, $z_j = z'_j = 0$, and $x_j = x'_j$. For $t < j \leq h$, $x_j = x'_j = 0$ and if $t < h$,

$$
(1 - 2\beta)z_j = I_j(x, z) = I_{\min}(x, z) = I_{\min}(x', z') = I_j(x', z') = (1 - 2\beta)z'_j
$$

so $z_j = z_j'$. Finally,
$$x_t = 1 - \sum_{j<t} x_j = 1 - \sum_{j<t} x_j' = x_t'$$

and
$$z_t = \begin{cases} \frac{1}{2}(1 - 2\sum_{t<j\leq h} z_j) = \frac{1}{2}(1 - 2\sum_{t<j\leq h} z_j') = z_t', & n \text{ even} \\ \frac{1}{2}(1 - 2\sum_{t<j<h} z_j - z_h) = \frac{1}{2}(1 - 2\sum_{t<j<h} z_j' - z_h) = z_t', & n \text{ odd} \end{cases}.$$

If $t = h$, then $z_t = 1 = z_t'$. Thus $(x, z) = (x', z')$.

Thus there is only one optimal basic feasible solution to Problem 12, and thus only one optimal solution by the fundamental theorem of linear programming. □

**Lemma 15.** *Let $(x, z)$ be the optimal solution to Problem 12, and let $t$ be the pivot element of $(x, z)$; suppose that $t \neq \frac{n+1}{2}$. Define*
$$L_t := \sum_{j<t} \frac{1}{q_j}.$$

*Then*
$$\lambda = \frac{2\beta q_t + \frac{1}{2}(1 - 2\beta)}{1 + q_t L_t + \frac{1}{2}(n - 2t)},$$

$$z_j = \begin{cases} 0, & j < t \\ \frac{1}{2}\left(1 - (n - 2t)\frac{\lambda}{1-2\beta}\right), & j \in \{t, n - t + 1\} \\ \frac{\lambda}{1-2\beta}, & t < j < n - t + 1 \end{cases}.$$

*Finally,*
$$x_j = \begin{cases} \frac{\lambda}{2\beta q_j}, & j < t \\ 1 - \frac{\lambda}{2\beta} L_t, & j = t \\ 0, & j > t \end{cases}.$$

*If $t = \frac{n+1}{2}$, then $x$ remains the same, but $z_t = 1$ and*
$$\lambda = \frac{2\beta q_t + (1 - 2\beta)}{1 + q_t L_t}.$$

*Proof.* For all $j < t$, $\lambda = 2\beta q_j x_j$, so $x_j = \frac{\lambda}{2\beta q_j}$. Moreover,
$$x_t = 1 - \sum_{j<t} x_j = 1 - \sum_{j<t} \frac{\lambda}{2\beta q_j} = 1 - \frac{\lambda}{2\beta} L_t.$$

Suppose that $n$ is odd and that $t = h = \frac{n+1}{2}$. Then $z_t = z_h$ must be 1 since for all $j < h$ $z_j = 0$. Moreover,
$$\lambda = 2\beta q_t x_t + (1 - 2\beta)z_t = 2\beta q_t \left(1 - \frac{\lambda}{2\beta} L_t\right) + (1 - 2\beta)$$

so, rearranging,
$$\lambda = \frac{2\beta q_t + (1 - 2\beta)}{1 + q_t L_t}.$$

Otherwise, $t < j \leq h$, $\lambda = (1 - 2\beta)z_j$, so $z_j = \frac{\lambda}{1-2\beta}$. Then since $\sum_{j\leq h} z_j + \sum_{h<j\leq n} z_{n-j+1} = 1$,

$$\begin{aligned} z_t &= \frac{1}{2}\left(1 - \sum_{j<t} z_j - \sum_{t<j\leq h} z_j - \sum_{h<j<n-t+1} z_{n-j+1} - \sum_{j>n-t+1} z_{n-j+1}\right) \\ &= \frac{1}{2}\left(1 - \sum_{t<j\leq h} z_j - \sum_{h<j<n-t+1} z_{n-j+1}\right) & (z_j = 0 \text{ for } j < t) \\ &= \frac{1}{2}\left(1 - (n - 2t)\frac{\lambda}{1 - 2\beta}\right). \end{aligned}$$

Moreover,

$$\lambda = 2\beta q_t x_t + (1 - 2\beta) z_t = 2\beta q_t \left(1 - \frac{\lambda}{2\beta} L_t\right) + (1 - 2\beta)\frac{1}{2}\left(1 - (n - 2t)\frac{\lambda}{1 - 2\beta}\right).$$

Rearranging,

$$\lambda(1 + q_t L_t + \tfrac{1}{2}(n - 2t)) = 2\beta q_t + \tfrac{1}{2}(1 - 2\beta),$$

so

$$\lambda = \frac{2\beta q_t + \tfrac{1}{2}(1 - 2\beta)}{1 + q_t L_t + \tfrac{1}{2}(n - 2t)}.$$

$\square$

