# OpenReview forum: "User-item fairness tradeoffs in recommendations"
_NeurIPS.cc/2024/Conference — NeurIPS 2024 poster_

### Official Review · Reviewer_t7QC · 2024-07-07

**Soundness:** 2
**Presentation:** 2
**Contribution:** 1
**Rating:** 3
**Confidence:** 4

**Summary:**

In this paper, the authors studied the tradeoff between user and item fairness in a recommendation setting. They proposed a constrained optimization problem that imposes user fairness as its objective and incorporates item fairness as its constraints. The authors also identified that (1) when user preferences are diverse, item fairness can be easily achieved; (2) when there is mis-estimation of user preference, imposing item constraint can lead to further cost for the users. Finally, the authors illustrated their findings using arXiv data.

**Strengths:**

- The problem of multi-sided fairness is an important problem, but most existing literature focus on studying single-sided fairness. Understanding the price of fairness is a meaningful problem for decision-makers in practice.

**Weaknesses:**

1. The model and framework used in this paper would require significantly more justification. More specifically,
- It is unclear why the authors choose to consider an optimization problem which maximizes normalized user preference as the objective subject to item fairness constraints. To me this is a rather counterintuitive choice and requires more motivation. Why not solving a dual-objective problem and treating user/item fairness in the same fashion? Alternatively, why not maximizing online platform's recommendation quality?
- The assumption that user and item share the same utility $w_{i,j}$ is too strong and can hardly hold in practice. In recommendation systems, a number of factors such as pricing, rankings, utility models, etc., could impact the item/user utilities in different ways. (See prior works such as [10, 11, 32], all of which use different definitions of user/item utilities.)
- The fairness notion that the authors adopt for users/items resemble a min-max type of fairness notion. This is again quite restrictive and requires more justification. Does your framework and results hold under alternative notions?

2. The theoretical results of this paper do not have sufficient technical contributions. For example,
- Proposition 2 basically uses the properties of BFS in LPs.
- Theorems 3 and 4 are only shown on a restrictive example (where there are only 2-3 types of users with opposing preferences in a pre-defined form). This raises the question of whether the theoretical results/insights can be extended to more general setups. Due to the restrictive setup and assumptions, the insights might also not have much practical relevance.

3. The price of fairness of multi-sided recommendations is a topic already studied in prior works. The insights provided in this work are not particularly surprising, nor distinguishable from prior works.
- For example, in [11] the authors also studied price of fairness and show that the price relates to the misalignment of platform/item/user objectives and designed an algorithm that resolves the issue of having unknown user/item data.
- The phenomena described in this work, such as more diverse user preferences naturally inducing item fairness, is also rather natural especially when min-max fairness is imposed, which encourages uniform item exposure. I'd expect such statement would fail to work if a different type of item fairness is considered.

4. No algorithm has been studied or introduced in this work. The empirical study of arXiv data merely serves to evaluate Problem (1) on arXiv data, but the model and framework itself already raises questions and requires extensive justification.

Overall, I think this work would require significantly more work in both its model and results and would recommend rejection.

**Questions:**

See weaknesses.

**Limitations:**

See weaknesses.

---

> ### Comment · Reviewer_yH5N · 2024-08-06
>
> Dear reviewer:
>
> I am another reviewer and an engineer who work in industrial RS. I would like to answer your question.
> "Why not solving a dual-objective problem and treating user/item fairness in the same fashion? Alternatively, why not maximizing online platform's recommendation quality?"
>
> In recommendation system, the key goal is to serve most user the item she want. The "platform's recommendation quality" could be evaluated in two perspectives: 1) is the users as a whole group satisfied? namely  the "user utilities" defined in line 135. 2) is every individual satisfied? In other world, maximize the user utilities of the least satisfied user, ie, "minimum user utilities" defined in this paper.
>
> In industrial system, we care about both the overall recommendation quality (point 1) and the recommendation quality for each user (point 2). Actually, if the least satisfied user is satisfied, the overall recommendation quality should be ok, although not optimal.
>
> While whether an item get enough chance to be seen, is not the key concern of commercial RS. Bad item should naturally be less shown to users. Thus, "treating user/item fairness in the same fashion" does not make sense. A reasonable formulation is to treat recommendation quality as objective, while item fairness as constrain.

---

> ### Author Rebuttal · Authors · 2024-08-07
>
> Thank you for your valuable feedback! We hope that the additional experiments we describe in the main rebuttal as well as the justification provided below address your concerns.
>
> ### Constrained optimization justification
>
> We agree that we could have modeled this as a dual objective problem – constraints at varying strength versus objectives with different weights are often interchangeable. We believe that our choice is reasonable, and using fairness constraints rather than objectives is consistent with related work [11]. We agree with reviewer yH5N that user and item fairness are not necessarily equally important in practice, and note that this could be captured by the dual-objective framework suggested by reviewer t7QC by using different weights on each objective.
>
> We will update our text to clarify this justification.
>
> ### Symmetric utility assumption
> Thank you for this feedback!
> - In the main response we provide experiments showing the robustness of our diversity result to alternative item utility models.
> - We also provide additional justification for why it is reasonable to restrict our analysis to this model: it is necessary and common in related work [10,32] to assume a specific utility model, and we believe a symmetric utility model captures a fundamental characteristic of producer preferences in many recommendation settings better than alternative models.
> We give more detail for these positions in the main response. We hope this addresses your concerns!
>
> ### Alternative notions of fairness
> We agree that other notions of fairness are also interesting, and re-ran our experiment in Figure 1(a) with Nash welfare as our definition of fairness for users and items, and see qualitatively the same results. We include an additional discussion of these points in the main response. We believe extending our theoretical results to additional definitions of fairness is an interesting question for future work.
>
> ### Insufficient technical contributions
> - We agree that Proposition 2 is mainly a result of using BFS properties: we included it as a separate result because it has an interesting qualitative interpretation.
> - However, we respectfully disagree that our work is not sufficiently technical – one challenge is to transform our problem into a setting where Proposition 2 can be applied, and then using the result for our conceptual findings. Proposition 1 provides a framework to transform the complicated program into a much simpler program. The proof involved manipulating the problem in a sequence of non-obvious ways. We then found closed form solutions to the transformed problem in Proposition 1, which was also non-trivial.
> We agree that the question of whether our results extend to populations with arbitrary preferences is crucial, and this was the intention of the experiments with arXiv data in Section 6. We hope that these experiments address your concerns, and agree that finding more general theoretical results is an interesting question for future work.
>
> ### Prior work
> Thank you for pointing this out!
> - [11] indeed defines the price of fairness, but in a different way from us: they examine the price of imposing fairness constraints on the revenue, thus capturing the impact of item/user fairness on revenue, while we examine the price of imposing item fairness constraints on user fairness, thus capturing the interplay between user and item fairness.
> - Moreover, their concept of objective misalignment (the difference in fairness between the {item, user} utility required by the constraints, and the {item, user} utility in a revenue-optimal solution) is not the same as our concept of user preference diversity (agreement between users' utilities). User preference diversity may *cause* objective alignment, but it is a different concept.
> - Their algorithm focuses on the algorithmic question of how to impose fairness constraints when preferences are unknown, but their analysis does not answer our question of whether fairness constraints disproportionately harm users with unknown preferences.
> We will update our related work section to make these points clearer.
>
> ### The diversity phenomenon is natural, and will fail with a different fairness definition.
> We agree that this phenomenon is natural, but disagree that it only would hold in a max-min fairness definition. Moreover, as mentioned above, we re-ran our arXiv experiment that demonstrated this result in the setting of Nash welfare fairness and saw the same effect (see Figure 3 in the PDF)
>
> ### No algorithm has been introduced
> Several algorithms have been developed to ensure multi-sided fairness in the existing literature [4,10,11]. The aim of this project is to understand factors affecting the trade-off between user fairness and item fairness. For example, in the algorithm in [11] the platform must choose the strength of the user- and item- fairness constraints. What constitutes a reasonable choice for the relative strength of the constraints depends heavily on how user and item fairness trade off, and is an important question independent from the development of the algorithm.
>
> Thank you for your consideration of our response!

---

### Official Review · Reviewer_rEVK · 2024-07-09

**Soundness:** 3
**Presentation:** 3
**Contribution:** 3
**Rating:** 7
**Confidence:** 3

**Summary:**

The paper works on the relationship between user fairness and item fairness in recommender system settings. A theoretical framework is proposed and some theoretical results and intuitions are provided based on the framework. The main results are the tradeoffs between fairness and 1) uncertainty and 2) diversity where theorems are proved together with some discussions.

---

I have read the rebuttal and other reviews. My rating is unchanged.

**Strengths:**

I like this paper. The theoretical framework is clear and useful. I expect more results can be derived out of this framework in the future e.g. assuming the utility w_{ij} is an estimator with certain bias and variance, or in practice one may not achieve the global optimal \rho^* but with a gap. The main results lie in theorem 3 and 4. Both results seem valid and intuitive. They illustrate how the price of fairness is changed when other factors like diversity/uncertainty are part of the consideration.

**Weaknesses:**

The empirical part on arxiv is quite light. I see more details in the appendix but the main body part needs more information to make it self-contained. In the theoretical framework part, I would make the setting more comprehensive but put additional assumptions in later sections in order to support those theorems e.g. I think the current setting is for a single-item recommendation problem and all users are independent. There will be correlation terms for different items when multiple items are shown to users together. In addition, the paper focuses on individual utility.

**Questions:**

I think the platform-level utility is missing from the discussion? In practice the platform controls the recommendation algorithm and therefore they may favor their utility as the primary objective.

The default for cold start user may not be the average of existing users, but selecting top-performing items. Suggest to consider this aspect in the revision process.

---

> ### Author Rebuttal · Authors · 2024-08-07
>
> Thank you for your thoughtful feedback and suggestions; we are glad you like our theoretical framework! We agree that there are likely to be other applications of the framework, and we are currently looking at ways to further draw out and interpret the sparsity result in the writing.
>
>
> ### Empirical details
>
> We will move more details from the appendix into the main body as you suggest for a more self-contained description.
>
>
> ### Theoretical framework generality
>
> Thank you for your comments regarding the theory writing. We agree that one possible writing approach is to introduce a general model and then later introduce the necessary assumptions. We will consider such a rewrite, though also would like to make our assumptions clear early.
>
>
> ###  Platform-level utility
>
> Our main goal was to understand the interplay between user and item fairness. We agree that platform utility  is a key follow-up question and include this in our list of suggested future work; this question is partially explored in [11], where they look at how the platform’s revenue trades off with fairness. We expect our diversity and mis-estimation results to extend to this setting. For example, with diverse users, giving every user their favorite item is optimal for users, items, and likely platform revenue as well.
>
>
> ### Cold start user default
>
> This is an interesting suggestion. We will try to model this and see what insights our framework gives for this case. We note that this choice partially occurs with our current framework without fairness constraints, since items that are generally popular will have high expected utility.
>
> Thank you for your consideration of our response!

---

> > ### Comment · Reviewer_rEVK · 2024-08-13
> > **Thanks**
> >
> > I have read the rebuttal and other reviews. Still, I think this is a solid work and will leave my rating (7-accept) unchanged

---

### Official Review · Reviewer_3QdJ · 2024-07-12

**Soundness:** 3
**Presentation:** 2
**Contribution:** 3
**Rating:** 6
**Confidence:** 3

**Summary:**

This paper investigates the trade-off between user fairness and item fairness in recommender systems. The authors develop a theoretical framework to characterize the user-item fairness trade-off by analyzing the recommendation strategy optimization problem. The following phenomena are found: 1. The more diversified the user preferences are, the smaller the user-item fairness trade-off is. 2. Inaccurate estimation of user preferences exacerbates the fairness trade-off, especially for new users. 3. In real data, moderate item fairness constraints have a small effect on user fairness but very strong constraints can significantly reduce user fairness.  Overall, the theoretical derivation part of this study is brilliant and provides theoretical assurance for the framework. It is a worthy study to explore the fairness problem of recommender systems in depth.

**Strengths:**

1. The trade-off relationship between user fairness and item fairness is systematically analyzed for the first time.
2. A new theoretical framework is proposed to simplify complex optimization problems.
3. the theoretical analysis is rigorous and provides in-depth mathematical proofs.

**Weaknesses:**

1. Limitations of the definition of fairness: the paper focuses mainly on minimized fairness indicators. Have other indicators of fairness been considered?
2 No specific methodology is given for actually achieving the three balances
3. The thesis assumes that the utility of users and items is symmetric, which may not always hold in reality.

**Questions:**

1. The paper assumes that the utility of users and items is symmetric, which may not always hold in reality. How does this assumption affect the generalisability of the results?
2. In this paper, fairness is quantified as the minimum normalized utility. This definition is supposed to follow Rawlsian fairness. it is not based on group fairness? fairness in this paper is measured using individuals. I am concerned about this because there are a large number of users in a recommender system and individual users may not be representative. I would like the authors to explain my concerns.
3. The selection of scenarios in the experimental section is limited, and the experiment on arXiv recommender systems, while meaningful, may not be sufficiently representative of all types of recommender systems. The generalisability of the results of this experiment could be discussed, or additional experiments in other domains could be considered.

**Limitations:**

1. Simplification of model assumptions: It assumes symmetric user and item utilities, which may not always hold in reality. It only considers the case of a single-item recommendation, whereas real systems usually recommend multiple items.
2. Limitations of the fairness definition: it focuses mainly on minimized fairness metrics and does not consider other possible fairness measures (e.g. group fairness).

---

> ### Author Rebuttal · Authors · 2024-08-07
>
> Thank you for your helpful feedback; we believe addressing them makes for a stronger paper.
>
> ### Fairness definitions
>
> We agree that it is important to study how our results extend to other definitions of fairness. In the main response, we give further justification for our choice, and show experiments demonstrating that our diversity result generalizes to Nash welfare fairness, $\sum_i \log(U_i(\rho))$.
>
> While still being an individual fairness measure (in considering fairness to users as individuals rather than as members of a certain group, which is outside the scope of our paper), Nash welfare incorporates a term from each user in the sum, resulting in a more holistic measure of user fairness. We hope this new experiment addresses your concern about representation and generalizability. We think that extending our results theoretically for additional definitions of fairness is an interesting question for future work.
>
> ### Achieving the three balances
>
> It is true that our paper does not provide a novel efficient algorithm for achieving multi-sided fairness (besides our reduced optimization problem); this is an interesting technical problem that has been explored in related work [4,10,11]. Our work complements these papers: we provide conceptual insights into the tradeoffs between user and item fairness. This is especially important for work like [11], whose algorithm needs to set hyper-parameters for the strength of user and item fairness constraints; setting these values appropriately requires understanding how and whether the two objectives trade off in the deployment setting.
>
> ### Symmetric utility
>
> In the main response, we provide experiments showing that the diversity phenomenon extends to settings where user and item utility are not symmetric, and further explain our reasoning for choosing this simplification.
>
> ### Single-item recommendation
>
> We agree that this is a limitation of our framework, and discuss this in the limitations section. Our intuition is that since in our framework the platform selects probabilistic policies, increasing the number of items will not affect the solutions qualitatively as much as a discrete policy would. We believe this is an important but difficult question for future work; one challenge with considering multiple items is modeling a tractable choice model that allows users to choose multiple items from the set of recommended items.
>
> ### ArXiv experiment generalizability
>
> Thank you for this suggestion; we will add a discussion of the generalizability of the experiment to other domains.
>
> Thank you for your consideration of this response!

---

> > ### Comment · Reviewer_3QdJ · 2024-08-14
> >
> > Thanks to the authors for the response. I keep my positive score.

---

### Official Review · Reviewer_yH5N · 2024-07-25

**Soundness:** 3
**Presentation:** 4
**Contribution:** 4
**Rating:** 7
**Confidence:** 3

**Summary:**

This paper develop a theoretical framework to analysis the trade-off between user fairness and item fairness. From the theoretically analysis, we understand that diverse user population benefits the recommendation, and users whose preferences are misestimated can be disadvantaged by the constraints on item fairness. The conclusion makes sense and is useful.

**Strengths:**

Clear definition on user fairness, item fairness, item utility constrained user utility, price etc. Good analysis on the two conclusion.

**Weaknesses:**

Relatively easy setting, e.g., only one item is recommended to a user. While the author honestly pointed out these weaknesses in the last paragraph, these weaknesses are acceptable.

**Questions:**

.

**Limitations:**

.

---

> ### Author Rebuttal · Authors · 2024-08-07
>
> Thank you for reading our work! We agree that only recommending one item is an aspect of our current theoretical model. We note that recommendations being probabilistic (e.g., one can think of an item being sampled each time period) somewhat mitigates this aspect, though agree that more explicitly modeling recommending multiple items (and how users select between them) is an important consideration for future work.

---

> > ### Comment · Reviewer_yH5N · 2024-08-12
> >
> > I have read the rebuttal

---

### Author Rebuttal · Authors · 2024-08-07

We thank the reviewers for their helpful feedback, and are glad that you found the paper “clear and useful” and the results to be both “rigorous” and “intuitive”. Multiple reviewers sought more justification for our **symmetry assumption** and **fairness definition**, so we discuss these issues here.

Overall, we appreciate the reviewer’s questions – there are always many ways to model this question, and the “correct” model often depends on the exact application. However, we believe that our primary insights are robust to the exact modeling choices, and that our choices are reasonable.

First, in response to the reviews, we include **additional arXiv experiments** in the response PDF.
- Figures 1 and 2 relax the **symmetric utility assumption** in the same setting as Figure 2 from the original paper, using exposure (and different levels of correlation) for item utility. The figures show that the diversity phenomenon is robust to this assumption, and that under different item utility models item fairness constraints still do not empirically increase the price of mis-estimation.
- Figure 3 shows that the diversity phenomenon is robust to **different definitions of fairness** by modifying the experiments of Figure 2 in the original paper to use **Nash welfare** [43] to define fairness. We also see the same phenomenon when we use the sum of the $k$ minimum participant utilities (which is a smoother extension of max-min fairness), with $k = 3$.

We will update our manuscript to include these experiments, showing that our results do not exactly depend on our theoretical choices.

Moreover, we believe that our restriction to symmetric utilities and minimized fairness for our theoretical analysis are reasonable choices for the theoretical analysis.

**Symmetric utility.** Our model assumes that users and items share a common utility $w_{ij}$ for recommending paper $j$ to user $i$.
1. **Extended model:** We would like to draw attention to line 1092 in Appendix E, where we provide an extension of our model in which user and item utilities are only assumed to be *proportional* to some shared value (that is, when item $j$ is recommended to user $i$, $i$ and $j$'s utilities are $\alpha_i w_{ij}$ and $\beta_j w_{ij}$ respectively for some $\alpha_i, \beta_j > 0$). Our theoretical results hold in this extended setting.
2. **Prior work:** Fixing a particular model of item utility is consistent with related work, as this is typically necessary for theoretical analysis. We use $w_{ij}^I = w_{ij}^U$, which resembles the "market share" utility model in [11]. *Exposure* ($w_{ij}^I = 1$ for all $i,j$) is also a popular choice [10,32] for this model, as we do in the new empirical analyses.
3. **Simple models:** As pointed out by the reviewers, neither model can fully capture the nuances of producer preferences. However, we believe that the symmetric utility model reflects a basic structure behind producer preferences in many cases ("items prefer to be recommended to users who like them, as that predicts consumption/purchase"), and that our theoretical findings should extend to settings with more or less this structure.
4. **Symmetry vs exposure:** We argue in Appendix E that having user and item preferences depend on a common value such as a purchase probability or click-through rate to be a more realistic representation of producer preferences than exposure in many cases. For example in an online marketplace a producer prefers their item to be recommended to users who are likely to buy their product, and in the paper recommendation setting an author prefers their paper to be recommended to a reader who is likely to engage with their work. This type of preference is not captured by exposure but is captured by the symmetric model.

We will update the writing to discuss these points, as well as alternative approaches.

**Minimized fairness.** Our model defines fairness as the normalized utility of the minimum user.
1. Min utility is a common notion of individual fairness in algorithmic fairness (called Rawlsian or egalitarian fairness); other reasonable choices include Nash welfare, which we now include in our empirical analysis ($\sum_i \log U_i(\rho)$, $\sum_j \log I_j(\rho)$ for users and items respectively; see pdf). We agree with reviewers that generalizing to other definitions is interesting and leave a theoretical extension for future work.

2. We expect the results to extend beyond this choice. One intuition for our diversity result is to consider the most user-diverse population: a population where every ranking of items is equally represented (and there is a consistent mapping from rankings to utilities). The user-optimal solution of giving every user their favorite item is also optimally item-fair for any reasonable* definition of individual item fairness, so there is no tradeoff. This intuition (and others) is independent of the definition of fairness, so we expect our results to generalize (and indeed they appear to do so in the new experiments in Figure 3).

*since it assigns all items the same utility and maximizes the total item utility.

Note: In this and the individual responses, we use the numbered references in our paper to refer to related work. For our experiments using different definitions of fairness, we add the following citation.
[43] Ioannis Caragiannis, David Kurokawa, Hervé Moulin, Ariel D. Procaccia, Nisarg Shah, and Junxing Wang. 2019. The Unreasonable Fairness of Maximum Nash Welfare. ACM Trans. Econ. Comput. 7, 3, Article 12 (August 2019), 32 pages.

---

### Decision · Program_Chairs · 2024-09-25

**Decision:**

Accept (poster)

**Comment:**

This paper investigates the trade-offs between user and item fairness in recommender systems, a crucial yet under-explored area. The authors present a novel theoretical framework and validate their findings empirically on the arXiv dataset, leading to valuable insights into fairness dynamics in recommendation.
Reviewers overall appreciate the paper's clarity, the importance of the problem tackled, and the rigor of the theoretical analysis. The identified phenomena, particularly the role of user diversity and the disproportionate impact of preference mis-estimation on fairness, are novel and hold significant practical implications for building fairer recommender systems. However, Reviewer t7QC raises several valid concerns regarding the paper's theoretical setup, specifically questioning the choice of fairness definition and the strong symmetry assumption in the utility model. While the authors have partially addressed these concerns by incorporating Nash Welfare in their empirical analysis and providing some justification for their modeling choices, a more thorough discussion and potential relaxation of these assumptions would significantly strengthen the paper.
Despite the limitations pointed out by Reviewer t7QC, the paper makes a valuable contribution to the field through the theoretical analysis and insights, and some practical relevance of the experiments. Therefore, the paper is recommended for acceptance. However, the authors are strongly encouraged to consider and address Reviewer t7QC's concerns in their final version of the paper -- specifically, discussing the limitations and exploring potential impact of alternative definitions on their findings, and providing a more thorough justification for the symmetry assumption in their utility model while discussing consequences of relaxing this assumption.